# Explanation Shift
# How Did the Distribution Shift Impact the Model?

## Abstract

The performance of machine learning models on new data is critical for their success in real-world applications. However, the model's performance may deteriorate if the new data is sampled from a different distribution than the training data. Current methods to detect shifts in the input or output data distributions have limitations in identifying model behavior changes. In this paper, we define *explanation shift* as the statistical comparison between how predictions from training data are explained and how predictions on new data are explained. We propose explanation shift as a key indicator to investigate the interaction between distribution shifts and learned models. We introduce an Explanation Shift Detector that operates on the explanation distributions, providing more sensitive and explainable changes in interactions between distribution shifts and learned models. We compare explanation shifts with other methods based on distribution shifts, showing that monitoring for explanation shifts results in more sensitive indicators for varying model behavior. We provide theoretical and experimental evidence and demonstrate the effectiveness of our approach on synthetic and real data. Additionally, we release an open-source Python package, `skshift`, which implements our method and provides usage tutorials for further reproducibility.

## 1  Introduction

ML theory provides means to forecast the quality of ML models on unseen data, provided that this data is sampled from the same distribution as the data used to train and evaluate the model. If unseen data is sampled from a different distribution than the training data, model quality may deteriorate, making monitoring how the model's behavior changes crucial.

Recent research has highlighted the impossibility of reliably estimating the performance of machine learning models on unseen data sampled from a different distribution in the absence of further assumptions about the nature of the shift [1, 2, 3]. State-of-the-art techniques attempt to model statistical distances between the distributions of the training and unseen data [4, 5] or the distributions of the model predictions [3, 6, 7]. However, these measures of *distribution shifts* only partially relate to changes of interaction between new data and trained models or they rely on the availability of a causal graph or types of shift assumptions, which limits their applicability. Thus, it is often necessary to go beyond detecting such changes and understand how the feature attribution changes [8, 9, 10, 4].

The field of explainable AI has emerged as a way to understand model decisions [11, 12] and interpret the inner workings of ML models [13]. The core idea of this paper is to go beyond the modeling of distribution shifts and monitor for *explanation shifts* to signal a change of interactions between learned models and dataset features in tabular data. We newly define explanation shift as the statistical comparison between how predictions from training data are explained and how predictions on new data are explained. In summary, our contributions are:

- We propose measures of explanation shifts as a key indicator for investigating the interaction between distribution shifts and learned models.
- We define an *Explanation Shift Detector* that operates on the explanation distributions allowing for more sensitive and explainable changes of interactions between distribution shifts and learned models.
- We compare our monitoring method that is based on explanation shifts with methods that are based on other kinds of distribution shifts. We find that monitoring for explanation shifts results in more sensitive indicators for varying model behavior.
- We release an open-source Python package `skshift`, which implements our "*Explanation Shift Detector*", along usage tutorials for reproducibility.

## 2 Foundations and Related Work

### 2.1 Basic Notions

Supervised machine learning induces a function $f_\theta : \text{dom}(X) \to \text{dom}(Y)$, from training data $\mathcal{D}^{tr} = \{(x_0^{tr}, y_0^{tr}) \dots, (x_n^{tr}, y_n^{tr})\}$. Thereby, $f_\theta$ is from a family of functions $f_\theta \in F$ and $\mathcal{D}^{tr}$ is sampled from the joint distribution $\mathbf{P}(X, Y)$ with predictor variables $X$ and target variable $Y$. $f_\theta$ is expected to generalize well on new, previously unseen data $\mathcal{D}_X^{new} = \{x_0^{new}, \dots, x_k^{new}\} \subseteq \text{dom}(X)$. We write $\mathcal{D}_X^{tr}$ to refer to $\{x_0^{tr}, \dots, x_n^{tr}\}$ and $\mathcal{D}_Y^{tr}$ to refer to $\mathcal{D}_Y^{tr} = \{y_0^{tr} \dots, y_n^{tr}\}$. For the purpose of formalizations and to define evaluation metrics, it is often convenient to assume that an oracle provides values $\mathcal{D}_Y^{new} = \{y_0^{new}, \dots, y_k^{new}\}$ such that $\mathcal{D}^{new} = \{(x_0^{new}, y_0^{new}), \dots, (x_k^{new}, y_k^{new})\} \subseteq \text{dom}(X) \times \text{dom}(Y)$.

The core machine learning assumption is that training data $\mathcal{D}^{tr}$ and novel data $\mathcal{D}^{new}$ are sampled from the same underlying distribution $\mathbf{P}(X, Y)$. The twin problems of *model monitoring* and recognizing that new data is *out-of-distribution* can now be described as predicting an absolute or relative performance drop between $\text{perf}(\mathcal{D}^{tr})$ and $\text{perf}(\mathcal{D}^{new})$, where $\text{perf}(\mathcal{D}) = \sum_{(x,y) \in \mathcal{D}} \ell_{\text{eval}}(f_\theta(x), y)$, $\ell_{\text{eval}}$ is a metric like 0-1-loss (accuracy), but $\mathcal{D}_Y^{new}$ is unknown and cannot be used for such judgment.

Therefore related work analyses distribution shifts between training and newly occurring data. Let two datasets $\mathcal{D}, \mathcal{D}'$ define two empirical distributions $\mathbf{P}(\mathcal{D}), \mathbf{P}(\mathcal{D}')$, then we write $\mathbf{P}(\mathcal{D}) \not\sim \mathbf{P}(\mathcal{D}')$ to express that $\mathbf{P}(\mathcal{D})$ is sampled from a different underlying distribution than $\mathbf{P}(\mathcal{D}')$ with high probability $p > 1 - \epsilon$ allowing us to formalize various types of distribution shifts.

**Definition 2.1** (Data Shift). We say that data shift occurs from $\mathcal{D}^{tr}$ to $\mathcal{D}_X^{new}$, if $\mathbf{P}(\mathcal{D}_X^{tr}) \not\sim \mathbf{P}(\mathcal{D}_X^{new})$.

Specific kinds of data shift are:

**Definition 2.2** (Univariate data shift). There is a univariate data shift between $\mathbf{P}(\mathcal{D}_X^{tr}) = \mathbf{P}(\mathcal{D}_{X_1}^{tr}, \dots, \mathcal{D}_{X_p}^{tr})$ and $\mathbf{P}(\mathcal{D}_X^{new}) = \mathbf{P}(\mathcal{D}_{X_1}^{new}, \dots, \mathcal{D}_{X_p}^{new})$, if $\exists i \in \{1 \dots p\} : \mathbf{P}(\mathcal{D}_{X_i}^{tr}) \not\sim \mathbf{P}(\mathcal{D}_{X_i}^{new})$.

**Definition 2.3** (Covariate data shift). There is a covariate data shift between $P(\mathcal{D}_X^{tr}) = \mathbf{P}(\mathcal{D}_{X_1}^{tr}, \dots, \mathcal{D}_{X_p}^{tr})$ and $\mathbf{P}(\mathcal{D}_X^{new}) = \mathbf{P}(\mathcal{D}_{X_1}^{new}, \dots, \mathcal{D}_{X_p}^{new})$ if $\mathbf{P}(\mathcal{D}_X^{tr}) \not\sim \mathbf{P}(\mathcal{D}_X^{new})$, which cannot only be caused by univariate shift.

The next two types of shift involve the interaction of data with the model $f_\theta$, which approximates the conditional $\frac{P(\mathcal{D}^{tr})}{P(\mathcal{D}_X^{tr})}$. Abusing notation, we write $f_\theta(\mathcal{D})$ to refer to the multiset $\{f_\theta(x) | x \in \mathcal{D}\}$.

**Definition 2.4** (Predictions Shift). There is a predictions shift between distributions $\mathbf{P}(\mathcal{D}_X^{tr})$ and $\mathbf{P}(\mathcal{D}_X^{new})$ related to model $f_\theta$ if $\mathbf{P}(f_\theta(\mathcal{D}_X^{tr})) \not\sim \mathbf{P}(f_\theta(\mathcal{D}_X^{new}))$.

**Definition 2.5** (Concept Shift). There is a concept shift between $\mathbf{P}(\mathcal{D}^{tr}) = P(\mathcal{D}_X^{tr}, \mathcal{D}_Y^{tr})$ and $\mathbf{P}(\mathcal{D}^{new}) = P(\mathcal{D}_X^{new}, \mathcal{D}_Y^{new})$ if conditional distributions change, i.e. $\frac{\mathbf{P}(\mathcal{D}^{tr})}{\mathbf{P}(\mathcal{D}_X^{tr})} \not\sim \frac{\mathbf{P}(\mathcal{D}^{new})}{\mathbf{P}(\mathcal{D}_X^{new})}$.

In practice, multiple types of shifts co-occur together and their disentangling may constitute a significant challenge that we do not address here [14, 15].

### 2.2 Related Work on Tabular Data

We briefly review the related works below. See Appendix A for a more detailed related work.

**Classifier two-sample test:** Evaluating how two distributions differ has been a widely studied topic in the statistics and statistical learning literature [16, 15, 17] and has advanced in recent years [18, 19, 20]. The use of supervised learning classifiers to measure statistical tests has been explored by Lopez-Paz et al. [21] proposing a classifier-based approach that returns test statistics to interpret differences between two distributions. We adopt their power test analysis and interpretability approach but apply it to the explanation distributions.

**Detecting distribution shift and its impact on model behaviour:** A lot of related work has aimed at detecting that data is from out-of-distribution. To this end, they have created several benchmarks that measure whether data comes from in-distribution or not [22, 23, 24, 25, 26]. In contrast, our main aim is to evaluate the impact of the distribution shift on the model.

A typical example is two-sample testing on the latent space such as described by Rabanser et al. [27]. However, many of the methods developed for detecting out-of-distribution data are specific to neural networks processing image and text data and can not be applied to traditional machine learning techniques. These methods often assume that the relationships between predictor and response variables remain unchanged, i.e., no concept shift occurs. Our work is applied to tabular data where techniques such as gradient boosting decision trees achieve state-of-the-art model performance [28, 29, 30].

**Impossibility of model monitoring:** Recent research findings have formalized the limitations of monitoring machine learning models in the absence of labelled data. Specifically [3, 31] prove the impossibility of predicting model degradation or detecting out-of-distribution data with certainty [32, 33, 34]. Although our approach does not overcome these limitations, it provides valuable insights for machine learning engineers to understand better changes in interactions resulting from shifting data distributions and learned models.

**Model monitoring and distribution shift under specific assumptions:** Under specific types of assumptions, model monitoring and distribution shift become feasible tasks. One type of assumption often found in the literature is to leverage causal knowledge to identify the drivers of distribution changes [35, 36, 37]. For example, Budhathoki et al. [35] use graphical causal models and feature attributions based on Shapley values to detect changes in the distribution. Similarly, other works aim to detect specific distribution shifts, such as covariate or concept shifts. Our approach does not rely on additional information, such as a causal graph, labelled test data, or specific types of distribution shift. Still, by the nature of pure concept shifts, the model behaviour remains unaffected and new data need to come with labelled responses to be detected.

**Explainability and distribution shift:** Lundberg et al. [38] applied Shapley values to identify possible bugs in the pipeline by visualizing univariate SHAP contributions. In our work we go beyond debugging and formalize the multivariate explanation distributions where we perform a two-sample classifier test to detect distribution shift impacts on the model. Furthermore, we provide a mathematical analysis of how the SHAP values contribute to detecting distribution shift.

### 2.3 Explainable AI: Local Feature Attributions

Attribution by Shapley values explains machine learning models by determining the relevance of features used by the model [38, 39]. The Shapley value is a concept from coalition game theory that aims to allocate the surplus generated by the grand coalition in a game to each of its players [40]. The Shapley value $\mathcal{S}_j$ for the $j$'th player is defined via a value function $\mathrm{val} : 2^N \to \mathbb{R}$ of players in $T$:

$$\mathcal{S}_j(\mathrm{val}) = \sum_{T \subseteq N \setminus \{j\}} \frac{|T|!(p - |T| - 1)!}{p!} (\mathrm{val}(T \cup \{j\}) - \mathrm{val}(T)) \tag{1}$$

In machine learning, $N = \{1, \ldots, p\}$ is the set of features occurring in the training data. Given that $x$ is the feature vector of the instance to be explained, and the term $\mathrm{val}_{f,x}(T)$ represents the prediction for the feature values in $T$ that are marginalized over features that are not included in $T$:

$$\mathrm{val}_{f,x}(T) = E_{X|X_T = x_T}[f(X)] - E_X[f(X)] \tag{2}$$

The Shapley value framework satisfies several theoretical properties [12, 40, 41, 42]. Our approach is based on the efficiency and uninformative properties:

**Efficiency Property.** Feature contributions add up to the difference of prediction from $x^\star$ and the expected value:

$$\sum_{j \in N} \mathcal{S}_j(f, x^\star) = f(x^\star) - E[f(X)]) \tag{3}$$

**Uninformativeness Property.** A feature $j$ that does not change the predicted value has a Shapley value of zero.

$$\forall x, x_j, x'_j : f(\{x_{N \setminus \{j\}}, x_j\}) = f(\{x_{N \setminus \{j\}}, x'_j\}) \Rightarrow \forall x : \mathcal{S}_j(f, x) = 0. \tag{4}$$

Our approach works with explanation techniques that fulfill efficiency and uninformative properties, and we use Shapley values as an example. It is essential to distinguish between the theoretical Shapley values and the different implementations that approximate them. We use TreeSHAP as an efficient implementation for tree-based models of Shapley values [38, 12, 43], mainly we use the observational (or path-dependent) estimation [44, 45, 46], and for linear models, we use the correlation dependent implementation that takes into account feature dependencies [47].

LIME is another explanation method candidate for out approach [48, 49]. LIME computes local feature attributions and also satisfies efficiency and uninformative properties, at least in theoretical aspects. However, the definition of neighborhoods in LIME and corresponding computational expenses impact its applicability. In Appendix F, we analyze LIME's relationship with Shapley values for the purpose of describing explanation shifts.

## 3 A Model for Explanation Shift Detection

Our model for explanation shift detection is sketched in Fig. 1. We define it step-by-step as follows:

**Definition 3.1** (Explanation distribution). An explanation function $\mathcal{S} : F \times \mathrm{dom}(X) \to \mathbb{R}^p$ maps a model $f_\theta$ and data $x \in \mathbb{R}^p$ to a vector of attributions $\mathcal{S}(f_\theta, x) \in \mathbb{R}^p$. We call $\mathcal{S}(f_\theta, x)$ an explanation. We write $\mathcal{S}(f_\theta, \mathcal{D})$ to refer to the empirical *explanation distribution* generated by $\{\mathcal{S}(f_\theta, x) | x \in \mathcal{D}\}$.

We use local feature attribution methods SHAP and LIME as explanation functions $\mathcal{S}$.

**Definition 3.2** (Explanation shift). Given a model $f_\theta$ learned from $\mathcal{D}^{tr}$, explanation shift with respect to the model $f_\theta$ occurs if $\mathcal{S}(f_\theta, \mathcal{D}_X^{new}) \not\sim \mathcal{S}(f_\theta, \mathcal{D}_X^{tr})$.

**Definition 3.3** (Explanation shift metrics). Given a measure of statistical distances $d$, explanation shift is measured as the distance between two explanations of the model $f_\theta$ by $d(\mathcal{S}(f_\theta, \mathcal{D}_X^{tr}), \mathcal{S}(f_\theta, \mathcal{D}_X^{new}))$.

We follow Lopez et al. [21] to define an explanation shift metrics based on a two-sample test classifier. We proceed as depicted in Figure 1. To counter overfitting, given the model $f_\theta$ trained on $\mathcal{D}^{tr}$, we compute explanations $\{\mathcal{S}(f_\theta, x) | x \in \mathcal{D}_X^{val}\}$ on an in-distribution validation data set $\mathcal{D}_X^{val}$. Given a dataset $\mathcal{D}_X^{new}$, for which the status of in- or out-of-distribution is unknown, we compute its explanations $\{\mathcal{S}(f_\theta, x) | x \in \mathcal{D}_X^{new}\}$. Then, we construct a two-samples dataset $E = \{(S(f_\theta, x), a_x) | x \in \mathcal{D}_X^{val}, a_x = 0\} \cup \{(S(f_\theta, x), a_x) | x \in \mathcal{D}_X^{new}, a_x = 1\}$ and we train a discrimination model $g_\psi : R^p \to \{0, 1\}$ on $E$, to predict if an explanation should be classified as in-distribution (ID) or out-of-distribution (OOD):

$$\psi = \arg\min_{\tilde{\psi}} \sum_{x \in \mathcal{D}_X^{val} \cup \mathcal{D}_X^{new}} \ell(g_{\tilde{\psi}}(\mathcal{S}(f_\theta, x)), a_x), \tag{5}$$

where $\ell$ is a classification loss function (e.g. cross-entropy). $g_\psi$ is our two-sample test classifier, based on which AUC yields a test statistic that measures the distance between the $D_X^{tr}$ explanations and the explanations of new data $D_X^{new}$.

Explanation shift detection allows us to detect *that* a novel dataset $D^{new}$ changes the model's behavior. Beyond recognizing explanation shift, using feature attributions for the model $g_\psi$, we can interpret *how* the features of the novel dataset $D_X^{new}$ interact differently with model $f_\theta$ than the features of the validation dataset $D_X^{val}$. These features are to be considered for model monitoring and for classifying new data as out-of-distribution.

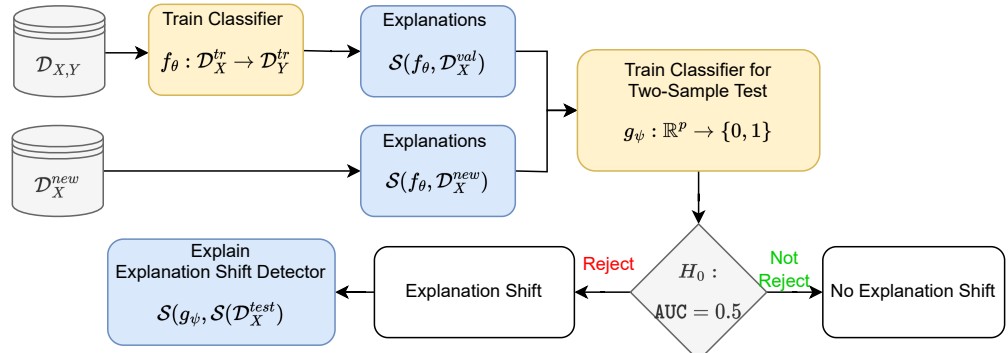

**Figure 1:** Our model for explanation shift detection. The model $f_\theta$ is trained on $\mathcal{D}^{tr}$ implying explanations for distributions $\mathcal{D}_X^{val}, \mathcal{D}_X^{new}$. The AUC of the two-sample test classifier $g_\psi$ decides for or against explanation shift. If an explanation shift occurred, it could be explained which features of the $\mathcal{D}_X^{new}$ deviated in $f_\theta$ compared to $\mathcal{D}_X^{val}$.

## 4 Relationships between Common Distribution Shifts and Explanation Shifts

This section analyses and compares data shifts, prediction shifts, with explanation shifts. Appendix B extends this analysis, and Appendix C draws from these analyses to derive experiments with synthetic data.

### 4.1 Explanation Shift vs Data Shift

One type of distribution shift that is challenging to detect comprises cases where the univariate distributions for each feature $j$ are equal between the source $\mathcal{D}_X^{tr}$ and the unseen dataset $\mathcal{D}_X^{new}$, but where interdependencies among different features change. Multi-covariance statistical testing is a hard taks with high sensitivity that can lead to false positives. The following example demonstrates that Shapley values account for co-variate interaction changes while a univariate statistical test will provide false negatives.

**Example 4.1.** *(Covariate Shift)* Let $D^{tr} \sim N\left(\begin{bmatrix} \mu_1 \\ \mu_2 \end{bmatrix}, \begin{bmatrix} \sigma_{X_1}^2 & 0 \\ 0 & \sigma_{X_2}^2 \end{bmatrix}\right) \times Y$. We fit a linear model $f_\theta(x_1, x_2) = \gamma + a \cdot x_1 + b \cdot x_2$. If $\mathcal{D}_X^{new} \sim N\left(\begin{bmatrix} \mu_1 \\ \mu_2 \end{bmatrix}, \begin{bmatrix} \sigma_{X_1}^2 & \rho\sigma_{X_1}\sigma_{X_2} \\ \rho\sigma_{X_1}\sigma_{X_2} & \sigma_{X_2}^2 \end{bmatrix}\right)$, then $\mathbf{P}(\mathcal{D}_{X_1}^{tr})$ and $\mathbf{P}(\mathcal{D}_{X_2}^{tr})$ are identically distributed with $\mathbf{P}(\mathcal{D}_{X_1}^{new})$ and $\mathbf{P}(\mathcal{D}_{X_2}^{new})$, respectively, while this does not hold for the corresponding $\mathcal{S}_j(f_\theta, \mathcal{D}_X^{tr})$ and $\mathcal{S}_j(f_\theta, \mathcal{D}_X^{new})$.

The detailed analysis of example 4.1 is given in Appendix B.2.

False positives frequently occur in out-of-distribution data detection when a statistical test recognizes differences between a source distribution and a new distribution, thought the differences do not affect the model behavior [28, 14]. Shapley values satisfy the *Uninformativeness* property, where a feature $j$ that does not change the predicted value has a Shapley value of 0 (equation 4).

**Example 4.2.** *Shifts on Uninformative Features.* Let the random variables $X_1, X_2$ be normally distributed with $N(0; 1)$. Let dataset $\mathcal{D}^{tr} \sim X_1 \times X_2 \times Y^{tr}$, with $Y^{tr} = X_1$. Thus $Y^{tr} \perp X_2$. Let $\mathcal{D}_X^{new} \sim X_1 \times X_2^{new}$ and $X_2^{new}$ be normally distributed with $N(\mu; \sigma^2)$ and $\mu, \sigma \in \mathbb{R}$. When $f_\theta$ is trained optimally on $\mathcal{D}^{tr}$ then $f_\theta(x) = x_1$. $\mathbf{P}(\mathcal{D}_{X_2})$ can be different from $\mathbf{P}(\mathcal{D}_{X_2}^{new})$ but $\mathcal{S}_2(f_\theta, \mathcal{D}_X^{tr}) = 0 = \mathcal{S}_2(f_\theta, \mathcal{D}_X^{new})$.

### 4.2 Explanation Shift vs Prediction Shift

Analyses of the explanations detect distribution shifts that interact with the model. In particular, if a prediction shift occurs, the explanations produced are also shifted.

**Proposition 1.** Given a model $f_\theta : \mathcal{D}_X \to \mathcal{D}_Y$. If $f_\theta(x') \neq f_\theta(x)$, then $\mathcal{S}(f_\theta, x') \neq \mathcal{S}(f_\theta, x)$.

By efficiency property of the Shapley values [47] (equation ((3))), if the prediction between two instances is different, then they differ in at least one component of their explanation vectors.

The opposite direction does not always hold:

**Example 4.3.** *(**Explanation shift not affecting prediction distribution**) Given $\mathcal{D}^{tr}$ is generated from $(X_1 \times X_2 \times Y)$, $X_1 \sim U(0,1)$, $X_2 \sim U(1,2)$, $Y = X_1 + X_2 + \epsilon$ and thus the optimal model is $f(x) = x_1 + x_2$. If $\mathcal{D}^{new}$ is generated from $X_1^{new} \sim U(1,2)$, $X_2^{new} \sim U(0,1)$, $Y^{new} = X_1^{new} + X_2^{new} + \epsilon$, the prediction distributions are identical $f_\theta(\mathcal{D}_X^{tr})$, $f_\theta(\mathcal{D}_X^{new}) \sim U(1,3)$, but explanation distributions are different $S(f_\theta, \mathcal{D}_X^{tr}) \not\sim S(f_\theta, \mathcal{D}_X^{new})$, because $\mathcal{S}_i(f_\theta, x) = \alpha_i \cdot x_i$.*

Thus, an explanation shift does not always imply a prediction shift.

### 4.3 Explanation Shift vs Concept Shift

Concept shift comprises cases where the covariates retain a given distribution, but their relationship with the target variable changes (cf. Section 2.1). This example shows the negative result that concept shift cannot be indicated by the detection of explanation shift.

**Example 4.4.** **Concept Shift** *Let $\mathcal{D}^{tr} \sim X_1 \times X_2 \times Y$, and create a synthetic target $y_i^{tr} = a_0 + a_1 \cdot x_{i,1} + a_2 \cdot x_{i,2} + \epsilon$. As new data we have $\mathcal{D}^{new} \sim X_1^{new} \times X_2^{new} \times Y$, with $y_i^{new} = b_0 + b_1 \cdot x_{i,1} + b_2 \cdot x_{i,2} + \epsilon$ whose coefficients are unknown at prediction stage. With coefficients $a_0 \neq b_0, a_1 \neq b_1, a_2 \neq b_2$. We train a linear regression $f_\theta : \mathcal{D}_X^{tr} \to \mathcal{D}_Y^{tr}$. Then explanations have the same distribution, $\mathbf{P}(\mathcal{S}(f_\theta, \mathcal{D}_X^{tr})) = \mathbf{P}(\mathcal{S}(f_\theta, \mathcal{D}_X^{new}))$, input data distribution $\mathbf{P}(\mathcal{D}_X^{tr}) = \mathbf{P}(\mathcal{D}_X^{new})$ and predictions $\mathbf{P}(f_\theta(\mathcal{D}_X^{tr})) = \mathbf{P}(f_\theta(\mathcal{D}_X^{new}))$. But there is no guarantee on the performance of $f_\theta$ on $\mathcal{D}_X^{new}$ [3]*

In general, concept shift cannot be detected because $\mathcal{D}_Y^{new}$ is unknown [3]. Some research studies have made specific assumptions about the conditional $\frac{P(\mathcal{D}^{new})}{P(\mathcal{D}_X^{new})}$ in order to monitor models and detect distribution shift [7, 50].

In Appendix B.2.2, we analyze a situation in which an oracle — hypothetically — provides $\mathcal{D}_Y^{new}$.

## 5 Empirical Evaluation

We perform core evaluations of explanation shift detection methods by systematically varying models $f$, model parametrizations $\theta$, and input data distributions $\mathcal{D}_X$. We complement core experiments described in this section by adding further experimental results in the appendix that (i) add details on experiments with synthetic data (Appendix C), (ii) add experiments on further natural datasets (Appendix D), (iii) exhibit a larger range of modeling choices (Appendix E), and (iv) include LIME as an explanation method (Appendix F). Core observations made in this section will only be confirmed and refined, but not countered in the appendix.

### 5.1 Baseline Methods and Datasets

**Baseline Methods.** We compare our method of explanation shift detection (Section 3) with several methods that aim to detect that input data is out-of-distribution: *(i)* statistical Kolmogorov Smirnov test on input data [27], *(ii)* classifier drift [51], *(iii)* prediction shift detection by Wasserstein distance [7], *(iv)* prediction shift detection by Kolmogorov-Smirnov test[4], and *(v)* model agnostic uncertainty estimation [10, 52]. Distribution Shift Metrics are scaled between 0 and 1. We also compare against Classifier Two-Sample Test [21] on different distributions as discussed in Section 4, viz. (vi) classifier two-sample test on input distributions ($g_\phi$) and (vii) classifier two-sample test on the predictions distributions ($g_\Upsilon$):

$$\phi = \arg\min_{\tilde{\phi}} \sum_{x \in \mathcal{D}_X^{val} \cup \mathcal{D}_X^{new}} \ell(g_{\tilde{\phi}}(x)), a_x) \qquad \Upsilon = \arg\min_{\tilde{\Upsilon}} \sum_{x \in \mathcal{D}_X^{val} \cup \mathcal{D}_X^{new}} \ell(g_{\tilde{\Upsilon}}(f_\theta(x)), a_x) \quad (6)$$

**Datasets.** In the main body of the paper we base our comparisons on the UCI Adult Income dataset [53] and on synthetic data. In the Appendix, we extend experiments to several other datasets, which confirm our findings: ACS Travel Time [54], ACS Employment [54], Stackoverflow dataset [55].

## 5.2 Experiments on Synthetic Data

Our first experiment on synthetic data showcases the two main contributions of our method: $(i)$ being more sensitive than prediction shift and input shift to changes in the model and $(ii)$ accounting for its drivers. We first generate a synthetic dataset with a shift similar to the multivariate shift one (cf. Section 4.2). However, we add an extra variable $X_3 = N(0,1)$ and generate our target $Y = X_1 \cdot X_2 + X_3$, and parametrize the multivariate shift between $\rho = r(X_1, X_2)$. We train the $f_\theta$ on $\mathcal{D}^{tr}$ using a gradient boosting decision tree, while for $g_\psi : \mathcal{S}(f_\theta, \mathcal{D}_X^{val}) \to \{0,1\}$, we use a logistic regression for both experiments. In Appendix E we benchmark other estimators and detectors.

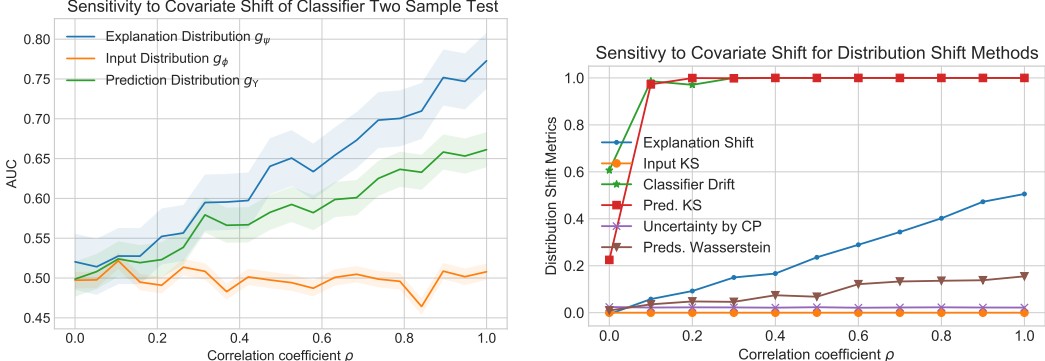

**Figure 2:** In the left figure, we apply the Classifier Two-Sample Test on (i) explanation distribution, (ii) input distribution, (iii) prediction distribution. Explanation distribution shows highest sensitivity. Comparison of the sensitivity of the *Explanation Shift Detector*. The right figure, related work comparison of distribution shift methods, good indicators should follow a progressive steady positive slope, following the correlation coefficient $\rho$.

Table 1 and Figure 2 show the results of our approach when learning on different distributions. In our sensitivity experiment, we observed that using the explanation shift led to higher sensitivity towards detecting distribution shift. This is due to the efficiency property of the Shapley values, which decompose $f_\theta(\mathcal{D}_X)$ into $\mathcal{S}(f_\theta, \mathcal{D}_X)$. Moreover, we can identify the features that are causing the drift by extracting the coefficients of $g_\psi$, providing global and local explainability.

The right image in Figure 2 compares our approach against Classifier Two Sample Testing for detecting multi-covariate shifts on different distributions. We can see how the explanations distributions have more sensitivity to the others. On the left image, the same experiment against other out-of-distribution detection methods such statistical differences on the input data (Input KS, Classifier Drift)[51, 4], which are model-independent; uncertainty estimation methods[52, 10, 56], whose effectiveness under specific types of shift is unclear; and statistical changes on the prediction distribution (K-S and Wasserstein Distance) [57, 58, 7], which can detect changes in model but lack sensitivity and accountability of the explanation shift. All metrics produce output scaled between 0 and 1.

**Table 1:** Conceptual comparison table over different detection methods over the examples discussed above. Learning a Classifier Two-Sample test $g$ over the explanation distributions is the only method that achieves the desired results and is accountable. We evaluate accountability by checking if the feature attributions of the detection method correspond with the synthetic shift generated in both scenarios

| Detection Method | Covariate | Uninformative | Accountability |
|---|:---:|:---:|:---:|
| Explanation distribution $(g_\psi)$ | ✓ | ✓ | ✓ |
| Input distribution$(g_\phi)$ | ✓ | ✗ | ✗ |
| Prediction distribution$(g_\Upsilon)$ | ✓ | ✓ | ✗ |
| Input KS | ✗ | ✗ | ✗ |
| Classifier Drift | ✓ | ✗ | ✗ |
| Output KS | ✓ | ✓ | ✗ |
| Output Wasserstein | ✓ | ✓ | ✗ |
| Uncertainty | $\sim$ | ✓ | ✓ |

## 5.3 Experiments on Natural Data: Inspecting Explanation Shifts

In the following experiments, we will provide use cases of our approach in two scenarios with natural data: $(i)$ novel group distribution shift and $(ii)$ geopolitical and temporal shift.

### 5.3.1 Novel Covariate Group

The distribution shift in this experimental set-up relies on the appearance of a new unseen group at the prediction stage (the group feature is not present in the covariates). We vary the ratio of presence of this unseen group in $\mathcal{D}_X^{new}$ data. As estimators, we use a gradient-boosting decision tree and a logistic regression(just when indicated); we use a logistic regression for the detector. We compare different estimators and detectors' performance in AppendixE.1 for a benchmark and Appendix E.2 for experiments varying hyperparameters.

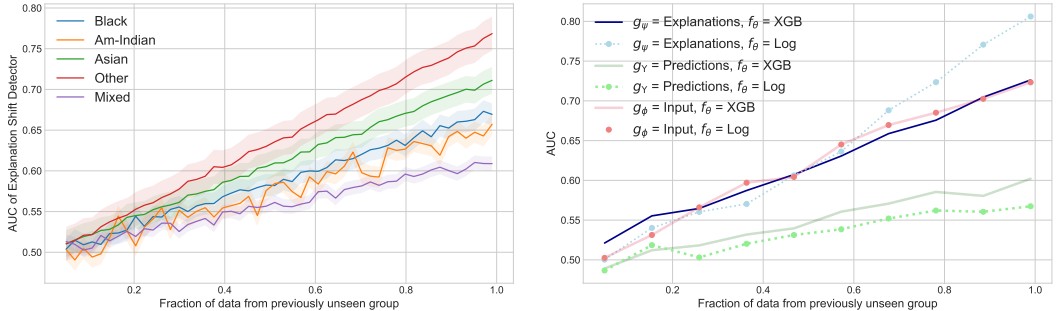

**Figure 3:** Novel group shift experiment on the UCI Adult Income dataset. Sensitivity (AUC) increases with the growing fraction of previously unseen social groups. Left figure: The explanation shift indicates that different social groups exhibit varying deviations from the distribution on which the model was trained. Right figure: We vary the model $f_\theta$ to be trained by XGBoost (solid lines) and Logistic Regression (dots), and the model $g$ to be trained on different distributions.

### 5.3.2 Geopolitical and Temporal Shift

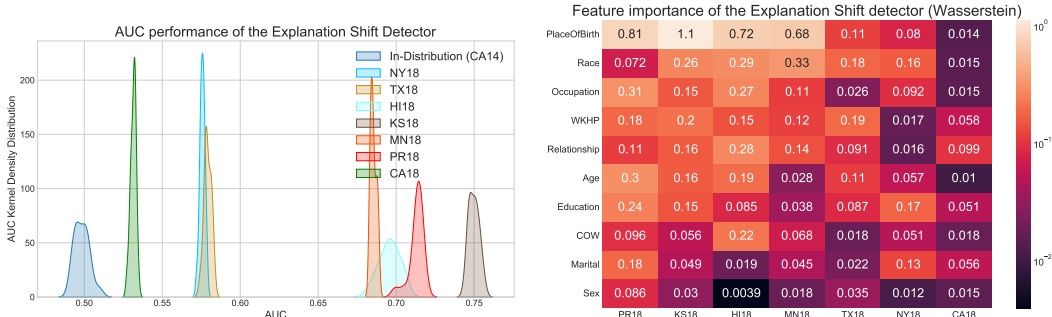

**Figure 4:** In the left figure, comparison of the performance of *Explanation Shift Detector*, in different states. In the right figure, strength analysis of features driving the change in the model, in the y-axis the features and on the x-axis the different states. Explanation shifts allow us to identify how the distribution shift of different features impacted the model.

In this section, we tackle a geopolitical and temporal distribution shift, for this, we train the model $f_\theta$ in California in 2014 and evaluate it in the rest of the states in 2018. The model $g_\theta$ is trained each time on each state using only the $\mathcal{D}_X^{new}$ in the absence of the label, and a 50/50 random train-test split evaluates its performance. As models, we use a gradient boosting decision tree[59, 60] as estimator $f_\theta$, and using logistic regression for the *Explanation Shift Detector*.

We hypothesize that the AUC of the "Explanation Shift Detector" on new data will be distinct from on ID data due to the OOD model explanations. Figure 4 illustrates the performance of our method on different data distributions, where the baseline is a hold-out set of $ID - CA14$. The AUC for

$CA18$, where there is only a temporal shift, is the closest to the baseline, and the OOD detection performance is better in the rest of the states. The most disparate state is Puerto Rico (PR18).

Our next objective is to identify the features where the explanations differ between $\mathcal{D}_X^{tr}$ and $\mathcal{D}_X^{new}$ data. To achieve this, we compare the distribution of linear coefficients of the detector between ID and New data. We use the Wasserstein distance as a distance measure, where we generate 1000 in-distribution bootstraps using a $63.2\%$ sampling fraction from California-14 and 1000 bootstraps from other states in 2018. In the right image of Figure 4, we observe that for PR18, the most crucial feature is the citizenship status[1].

Furthermore, we conduct an across-task evaluation by comparing the performance of the "Explanation Shift Detector" on another prediction task in the Appendix D. Although some features are present in both prediction tasks, the weights and importance order assigned by the "Explanation Shift Detector" differ. One of this method's advantages is that it identifies differences in distributions and how they relate to the model.

## 6 Discussion

In this study, we conducted a comprehensive evaluation of explanation shift by systematically varying models ($f$), model parametrizations ($\theta$), feature attribution explanations ($\mathcal{S}$), and input data distributions ($\mathcal{D}_X$). Our objective was to investigate the impact of distribution shift on the model by explanation shift and gain insights into its characteristics and implications.

Our approach cannot detect concept shifts, as concept shift requires understanding the interaction between prediction and response variables. By the nature of pure concept shifts, such changes do not affect the model. To be understood, new data need to come with labelled responses. We work under the assumption that such labels are not available for new data, nor do we make other assumptions; therefore, our method is not able to predict the degradation of prediction performance under distribution shifts. All papers such as [3, 10, 61, 31, 32, 62, 7] that address the monitoring of prediction performance have the same limitation. Only under specific assumptions, e.g., no occurrence of concept shift or causal graph availability, can performance degradation be predicted with reasonable reliability.

The potential utility of explanation shifts as distribution shift indicators that affect the model in computer vision or natural language processing tasks remains an open question. We have used Shapley values to derive indications of explanation shifts, but other AI explanation techniques may be applicable and come with their advantages.

## 7 Conclusions

Commonly, the problem of detecting the impact of the distribution shift on the model has relied on measurements for detecting shifts in the input or output data distributions or relied on assumptions either on the type of distribution shift or causal graphs availability. In this paper, we have provided evidence that explanation shifts can be a more suitable indicator for detecting and identifying distribution shifts' impact in machine learning models. We provide software, mathematical analysis examples, synthetic data, and real-data experimental evaluation. We found that measures of explanation shift can provide more insights than input distribution and prediction shift measures when monitoring machine learning models.

**Reproducibility Statement**

To ensure reproducibility, we make the data, code repositories, and experiments publicly available [2]. Also, an open-source Python package `skshift`[3] is attached with methods routines and tutorials. For our experiments, we used default `scikit-learn` parameters [63]. We describe the system requirements and software dependencies of our experiments. Experiments were run on a 4 vCPU server with 32 GB RAM.

---

[1]The ACS PUMS data dictionary contains a comprehensive list of available variables https://www.census.gov/programs-surveys/acs/microdata/documentation.html

[2]https://anonymous.4open.science/r/ExplanationShift-C0C0/README.md

[3]https://anonymous.4open.science/r/skshift-65A5/README.md

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

# Contents

## A  Extended Related Work

This section provides an in-depth review of the related theoretical works that inform our research.

### A.1  Out-Of-Distribution Detection

Evaluating how two distributions differ has been a widely studied topic in the statistics and statistical learning literature [16, 15, 17], that have advanced recently in last years [18, 19, 20]. [27] provides a comprehensive empirical investigation, examining how dimensionality reduction and two-sample testing might be combined to produce a practical pipeline for detecting distribution shifts in real-life machine learning systems. Other methods to detect if new data is OOD have relied on neural networks based on the prediction distributions [57, 58]. They use the maximum softmax probabilities/likelihood as a confidence score [64], temperature or energy-based scores [65, 66, 67], they extract information from the gradient space [68], they fit a Gaussian distribution to the embedding, or they use the Mahalanobis distance for out-of-distribution detection [19, 69].

Many of these methods are explicitly developed for neural networks that operate on image and text data, and often they can not be directly applied to traditional ML techniques. For image and text data, one may build on the assumption that the relationships between relevant predictor variables ($X$) and response variables ($Y$) remain unchanged, i.e., that no *concept shift* occurs. For instance, the essence of how a dog looks remains unchanged over different data sets, even if contexts may change. Thus, one can define invariances on the latent spaces of deep neural models, which are not applicable to tabular data in a likewise manner. For example, predicting buying behavior before, during, and after the COVID-19 pandemic constitutes a conceptual shift that is not amenable to such methods. We focus on such tabular data where techniques such as gradient boosting decision trees achieve state-of-the-art model performance [28, 29, 30].

### A.2  Explainability and Distribution Shift

Another approach using Shapley values by Balestra et al. [70] allows for tracking distributional shifts and their impact among for categorical time series using slidSHAP, a novel method for unlabelled data streams. In our work, we define the explanation distributions and exploit its theoretical properties under distribution shift where we perform a two-sample classifier test to detect

Haut et al. [71] track changes in the distribution of model parameter values that are directly related to the input features to identify concept drift early on in data streams. In a more recent paper,Haug et al. [9] also exploits the idea that local changes to feature attributions and distribution shifts are strongly intertwined and uses this idea to update the local feature attributions efficiently. Their work focuses on model retraining and concept shift, in our work the original estimator $f_\theta$ remains unaltered, and since we are in an unsupervised monitoring scenario we can't detect concept shift see discussion in Section 6

## B  Extended Analytical Examples

This appendix provides more details about the analytical examples presented in Section 4.1.

### B.1  Explanation Shift vs Prediction Shift

**Proposition 2.** Given a model $f_\theta : \mathcal{D}_X \to \mathcal{D}_Y$. If $f_\theta(x') \neq f_\theta(x)$, then $\mathcal{S}(f_\theta, x') \neq \mathcal{S}(f_\theta, x)$.

$$\text{Given} \quad f_\theta(x) \neq f_\theta(x') \tag{7}$$

$$\sum_{j=1}^{p} \mathcal{S}_j(f_\theta, x) = f_\theta(x) - E_X[f_\theta(\mathcal{D}_X)] \tag{8}$$

$$\text{then} \quad \mathcal{S}(f, x) \neq \mathcal{S}(f, x') \tag{9}$$

**Example B.1.** *Explanation shift that does not affect the prediction distribution* Given $\mathcal{D}^{tr}$ is *generated from* $(X_1, X_2, Y), X_1 \sim U(0, 1), X_2 \sim U(1, 2), Y = X_1 + X_2 + \epsilon$ *and thus the model is* $f(x) = x_1 + x_2$. *If* $\mathcal{D}^{new}$ *is generated from* $X_1^{new} \sim U(1, 2), X_2^{new} \sim U(0, 1)$, *the prediction distributions are identical* $f_\theta(\mathcal{D}_X^{tr}), f_\theta(\mathcal{D}_X^{new})$, *but explanation distributions are different* $S(f_\theta, \mathcal{D}_X^{tr}) \neq S(f_\theta, \mathcal{D}_X^{new})$

$$\forall i \in \{1, 2\} \quad \mathcal{S}_i(f_\theta, x) = \alpha_i \cdot x_i \tag{10}$$

$$\forall i \in \{1, 2\} \Rightarrow \mathcal{S}_i(f_\theta, \mathcal{D}_X)) \neq \mathcal{S}_i(f_\theta, \mathcal{D}_X^{new}) \tag{11}$$

$$\Rightarrow f_\theta(\mathcal{D}_X) = f_\theta(\mathcal{D}_X^{new}) \tag{12}$$

### B.2  Explanation Shifts vs Input Data Distribution Shifts

#### B.2.1  Multivariate Shift

**Example B.2.** *Multivariate Shift Let* $D_X^{tr} = (\mathcal{D}_{X_1}^{new}, \mathcal{D}_{X_2}^{new}) \sim N\left(\begin{bmatrix} \mu_1 \\ \mu_2 \end{bmatrix}, \begin{bmatrix} \sigma_{x_1}^2 & 0 \\ 0 & \sigma_{x_2}^2 \end{bmatrix}\right), \mathcal{D}_X^{new} = (\mathcal{D}_{X_1}^{new}, \mathcal{D}_{X_2}^{new}) \sim N\left(\begin{bmatrix} \mu_1 \\ \mu_2 \end{bmatrix}, \begin{bmatrix} \sigma_{x_1}^2 & \rho\sigma_{x_1}\sigma_{x_2} \\ \rho\sigma_{x_1}\sigma_{x_2} & \sigma_{x_2}^2 \end{bmatrix}\right)$. *We fit a linear model* $f_\theta(X_1, X_2) = \gamma + a \cdot X_1 + b \cdot X_2$. $\mathcal{D}_{X_1}$ *and* $\mathcal{D}_{X_2}$ *are identically distributed with* $\mathcal{D}_{X_1}^{new}$ *and* $\mathcal{D}_{X_2}^{new}$, *respectively, while this does not hold for the corresponding SHAP values* $\mathcal{S}_j(f_\theta, \mathcal{D}_X^{tr})$ *and* $\mathcal{S}_j(f_\theta, \mathcal{D}_X^{val})$.

$$\mathcal{S}_1(f_\theta, x) = a(x_1 - \mu_1) \tag{13}$$

$$\mathcal{S}_1(f_\theta, x^{new}) = \tag{14}$$

$$= \frac{1}{2}[\text{val}(\{1, 2\}) - \text{val}(\{2\})] + \frac{1}{2}[\text{val}(\{1\}) - \text{val}(\emptyset)] \tag{15}$$

$$\text{val}(\{1, 2\}) = E[f_\theta | X_1 = x_1, X_2 = x_2] = ax_1 + bx_2 \tag{16}$$

$$\text{val}(\emptyset) = E[f_\theta] = a\mu_1 + b\mu_2 \tag{17}$$

$$\text{val}(\{1\}) = E[f_\theta(x) | X_1 = x_1] + b\mu_2 \tag{18}$$

$$\text{val}(\{1\}) = \mu_1 + \rho\frac{\rho_{x_1}}{\sigma_{x_2}}(x_1 - \sigma_1) + b\mu_2 \tag{19}$$

$$\text{val}(\{2\}) = \mu_2 + \rho\frac{\sigma_{x_2}}{\sigma_{x_1}}(x_2 - \mu_2) + a\mu_1 \tag{20}$$

$$\Rightarrow \mathcal{S}_1(f_\theta, x^{new}) \neq a(x_1 - \mu_1) \tag{21}$$

#### B.2.2  Concept Shift

One of the most challenging types of distribution shift to detect are cases where distributions are equal between source and unseen data-set $\mathbf{P}(\mathcal{D}_X^{tr}) = \mathbf{P}(\mathcal{D}_X^{new})$ and the target variable $\mathbf{P}(\mathcal{D}_Y^{tr}) = \mathbf{P}(\mathcal{D}_Y^{new})$ and what changes are the relationships that features have with the target $\mathbf{P}(\mathcal{D}_Y^{tr}|\mathcal{D}_X^{tr}) \neq \mathbf{P}(\mathcal{D}_Y^{new}|\mathcal{D}_X^{new})$, this kind of distribution shift is also known as concept drift or posterior shift [14] and is especially difficult to notice, as it requires labeled data to detect. The following example

compares how the explanations change for two models fed with the same input data and different target relations.

**Example B.3.** *Concept shift Let $\mathcal{D}_X = (X_1, X_2) \sim N(\mu, I)$, and $\mathcal{D}_X^{new} = (X_1^{new}, X_2^{new}) \sim N(\mu, I)$, where $I$ is an identity matrix of order two and $\mu = (\mu_1, \mu_2)$. We now create two synthetic targets $Y = a + \alpha \cdot X_1 + \beta \cdot X_2 + \epsilon$ and $Y^{new} = a + \beta \cdot X_1 + \alpha \cdot X_2 + \epsilon$. Let $f_\theta$ be a linear regression model trained on $f_\theta : \mathcal{D}_X \to \mathcal{D}_Y$ and $h_\phi$ another linear model trained on $h_\phi : \mathcal{D}_X^{new} \to \mathcal{D}_Y^{new}$. Then $\mathbf{P}(f_\theta(X)) = \mathbf{P}(h_\phi(X^{new}))$, $P(X) = \mathbf{P}(X^{new})$ but $\mathcal{S}(f_\theta, X) \neq \mathcal{S}(h_\phi, X)$.*

$$X \sim N(\mu, \sigma^2 \cdot I), X^{new} \sim N(\mu, \sigma^2 \cdot I) \tag{22}$$

$$\to P(\mathcal{D}_X) = P(\mathcal{D}_X^{new}) \tag{23}$$

$$Y \sim a + \alpha N(\mu, \sigma^2) + \beta N(\mu, \sigma^2) + N(0, \sigma'^2) \tag{24}$$

$$Y^{new} \sim a + \beta N(\mu, \sigma^2) + \alpha N(\mu, \sigma^2) + N(0, \sigma'^2) \tag{25}$$

$$\to P(\mathcal{D}_Y) = P(\mathcal{D}_Y^{new}) \tag{26}$$

$$\mathcal{S}(f_\theta, \mathcal{D}_X) = \begin{pmatrix} \alpha(X_1 - \mu_1) \\ \beta(X_2 - \mu_2) \end{pmatrix} \sim \begin{pmatrix} N(\mu_1, \alpha^2\sigma^2) \\ N(\mu_2, \beta^2\sigma^2) \end{pmatrix} \tag{27}$$

$$\mathcal{S}(h_\phi, \mathcal{D}_X) = \begin{pmatrix} \beta(X_1 - \mu_1) \\ \alpha(X_2 - \mu_2) \end{pmatrix} \sim \begin{pmatrix} N(\mu_1, \beta^2\sigma^2) \\ N(\mu_2, \alpha^2\sigma^2) \end{pmatrix} \tag{28}$$

$$\text{If} \quad \alpha \neq \beta \to \mathcal{S}(f_\theta, \mathcal{D}_X) \neq \mathcal{S}(h_\phi, \mathcal{D}_X) \tag{29}$$

# C  Further Experiments on Synthetic Data

This experimental section explores the detection of distribution shift on the previous synthetic examples.

## C.1  Detecting multivariate shift

Given two bivariate normal distributions $\mathcal{D}_X = (X_1, X_2) \sim N\left(0, \begin{bmatrix} 1 & 0 \\ 0 & 1 \end{bmatrix}\right)$ and $\mathcal{D}_X^{new} = (X_1^{new}, X_2^{new}) \sim N\left(0, \begin{bmatrix} 1 & 0.2 \\ 0.2 & 1 \end{bmatrix}\right)$, then, for each feature $j$ the underlying distribution is equally distributed between $\mathcal{D}_X$ and $\mathcal{D}_X^{new}$, $\forall j \in \{1, 2\} : P(\mathcal{D}_{X_j}) = P(\mathcal{D}_{X_j}^{new})$, and what is different are the interaction terms between them. We now create a synthetic target $Y = X_1 \cdot X_2 + \epsilon$ with $\epsilon \sim N(0, 0.1)$ and fit a gradient boosting decision tree $f_\theta(\mathcal{D}_X)$. Then we compute the SHAP explanation values for $\mathcal{S}(f_\theta, \mathcal{D}_X)$ and $\mathcal{S}(f_\theta, \mathcal{D}_X^{new})$

**Table 2:** Displayed results are the one-tailed p-values of the Kolmogorov-Smirnov test comparison between two underlying distributions. Small p-values indicate that compared distributions would be very unlikely to be equally distributed. SHAP values correctly indicate the interaction changes that individual distribution comparisons cannot detect

| Comparison | p-value | Conclusions |
|---|---|---|
| $\mathbf{P}(\mathcal{D}_{X_1}), \mathbf{P}(\mathcal{D}_{X_1}^{new})$ | 0.33 | Not Distinct |
| $\mathbf{P}(\mathcal{D}_{X_2}), \mathbf{P}(\mathcal{D}_{X_2}^{new})$ | 0.60 | Not Distinct |
| $\mathcal{S}_1(f_\theta, \mathcal{D}_X), \mathcal{S}_1(f_\theta, \mathcal{D}_X^{new})$ | 3.9e−153 | Distinct |
| $\mathcal{S}_2(f_\theta, \mathcal{D}_X), \mathcal{S}_2(f_\theta, \mathcal{D}_X^{new})$ | 2.9e−148 | Distinct |

Having drawn $50,000$ samples from both $\mathcal{D}_X$ and $\mathcal{D}_X^{new}$, in Table 2, we evaluate whether changes in the input data distribution or on the explanations are able to detect changes in covariate distribution. For this, we compare the one-tailed p-values of the Kolmogorov-Smirnov test between the input data distribution and the explanations distribution. Explanation shift correctly detects the multivariate distribution change that univariate statistical testing can not detect.

## C.2 Detecting concept shift

As mentioned before, concept shift cannot be detected if new data comes without target labels. If new data is labelled, the explanation shift can still be a useful technique for detecting concept shifts.

Given a bivariate normal distribution $\mathcal{D}_X = (X_1, X_2) \sim N(1, I)$ where $I$ is an identity matrix of order two. We now create two synthetic targets $Y = X_1^2 \cdot X_2 + \epsilon$ and $Y^{new} = X_1 \cdot X_2^2 + \epsilon$ and fit two machine learning models $f_\theta : \mathcal{D}_X \to \mathcal{D}_Y)$ and $h_\Upsilon : \mathcal{D}_X \to \mathcal{D}_Y^{new})$. Now we compute the SHAP values for $\mathcal{S}(f_\theta, \mathcal{D}_X)$ and $\mathcal{S}(h_\Upsilon, \mathcal{D}_X)$

**Table 3:** Distribution comparison for synthetic concept shift. Displayed results are the one-tailed p-values of the Kolmogorov-Smirnov test comparison between two underlying distributions

| Comparison | Conclusions |
|---|---|
| $\mathbf{P}(\mathcal{D}_X), \mathbf{P}(\mathcal{D}_X^{new})$ | Not Distinct |
| $\mathbf{P}(\mathcal{D}_Y), \mathbf{P}(\mathcal{D}_Y^{new})$ | Not Distinct |
| $\mathbf{P}(f_\theta(\mathcal{D}_X)), \mathbf{P}(h_\Upsilon(\mathcal{D}_X^{new}))$ | Not Distinct |
| $\mathbf{P}(\mathcal{S}(f_\theta, \mathcal{D}_X)), \mathbf{P}(\mathcal{S}(h_\Upsilon, \mathcal{D}_X))$ | Distinct |

In Table 3, we see how the distribution shifts are not able to capture the change in the model behavior while the SHAP values are different. The "Distinct/Not distinct" conclusion is based on the one-tailed p-value of the Kolmogorov-Smirnov test with a $0.05$ threshold drawn out of $50,000$ samples for both distributions. As in the synthetic example, in table 3 SHAP values can detect a relational change between $\mathcal{D}_X$ and $\mathcal{D}_Y$, even if both distributions remain equivalent.

## C.3 Uninformative features on synthetic data

To have an applied use case of the synthetic example from the methodology section, we create a three-variate normal distribution $\mathcal{D}_X = (X_1, X_2, X_3) \sim N(0, I_3)$, where $I_3$ is an identity matrix of order three. The target variable is generated $Y = X_1 \cdot X_2 + \epsilon$ being independent of $X_3$. For both, training and test data, $50,000$ samples are drawn. Then out-of-distribution data is created by shifting $X_3$, which is independent of the target, on test data $\mathcal{D}_{X_3}^{new} = \mathcal{D}_{X_3}^{te} + 1$.

**Table 4:** Distribution comparison when modifying a random noise variable on test data. The input data shifts while explanations and predictions do not.

| Comparison | Conclusions |
|---|---|
| $\mathbf{P}(\mathcal{D}_{X_3}^{te}), \mathbf{P}(\mathcal{D}_{X_3}^{new})$ | Distinct |
| $f_\theta(\mathcal{D}_X^{te}), f_\theta(\mathcal{D}_X^{new})$ | Not Distinct |
| $\mathcal{S}(f_\theta, \mathcal{D}_X^{te}), \mathcal{S}(f_\theta, \mathcal{D}_X^{new})$ | Not Distinct |

In Table 4, we see how an unused feature has changed the input distribution, but the explanation distributions and performance evaluation metrics remain the same. The "Distinct/Not Distinct" conclusion is based on the one-tailed p-value of the Kolmogorov-Smirnov test drawn out of $50,000$ samples for both distributions.

## C.4 Explanation shift that does not affect the prediction

In this case we provide a situation when we have changes in the input data distributions that affect the model explanations but do not affect the model predictions due to positive and negative associations between the model predictions and the distributions cancel out producing a vanishing correlation in the mixture of the distribution (Yule's effect 4.2).

We create a train and test data by drawing $50,000$ samples from a bi-uniform distribution $X_1 \sim U(0, 1), \quad X_2 \sim U(1, 2)$ the target variable is generated by $Y = X_1 + X_2$ where we train our model $f_\theta$. Then if out-of-distribution data is sampled from $X_1^{new} \sim U(1, 2), X_2^{new} \sim U(0, 1)$

In Table 5, we see how an unused feature has changed the input distribution, but the explanation distributions and performance evaluation metrics remain the same. The "Distinct/Not Distinct" conclusion is based on the one-tailed p-value of the Kolmogorov-Smirnov test drawn out of $50,000$ samples for both distributions.

**Table 5:** Distribution comparison over how the change on the contributions of each feature can cancel out to produce an equal prediction (cf. Section 4.2), while explanation shift will detect this behaviour changes on the predictions will not.

| Comparison | Conclusions |
|---|---|
| $f(\mathcal{D}_X^{te}), f(\mathcal{D}_X^{new})$ | Not Distinct |
| $\mathcal{S}(f_\theta, \mathcal{D}_{X_2}^{te}), \mathcal{S}(f_\theta, \mathcal{D}_{X_2}^{new})$ | Distinct |
| $\mathcal{S}(f_\theta, \mathcal{D}_{X_1}^{te}), \mathcal{S}(f_\theta, \mathcal{D}_{X_1}^{new})$ | Distinct |

# D  Further Experiments on Real Data

In this section, we extend the prediction task of the main body of the paper. The methodology used follows the same structure. We start by creating a distribution shift by training the model $f_\theta$ in California in 2014 and evaluating it in the rest of the states in 2018, creating a geopolitical and temporal shift. The model $g_\theta$ is trained each time on each state using only the $X^{New}$ in the absence of the label, and its performance is evaluated by a 50/50 random train-test split. As models, we use a gradient boosting decision tree[59, 60] as estimator $f_\theta$, approximating the Shapley values by TreeExplainer [38], and using logistic regression for the *Explanation Shift Detector*.

## D.1  ACS Employment

The objective of this task is to determine whether an individual aged between 16 and 90 years is employed or not. The model's performance was evaluated using the AUC metric in different states, except PR18, where the model showed an explanation shift. The explanation shift was observed to be influenced by features such as Citizenship and Military Service. The performance of the model was found to be consistent across most of the states, with an AUC below 0.60. The impact of features such as difficulties in hearing or seeing was negligible in the distribution shift impact on the model. The left figure in Figure 5 compares the performance of the Explanation Shift Detector in different states for the ACS Employment dataset.

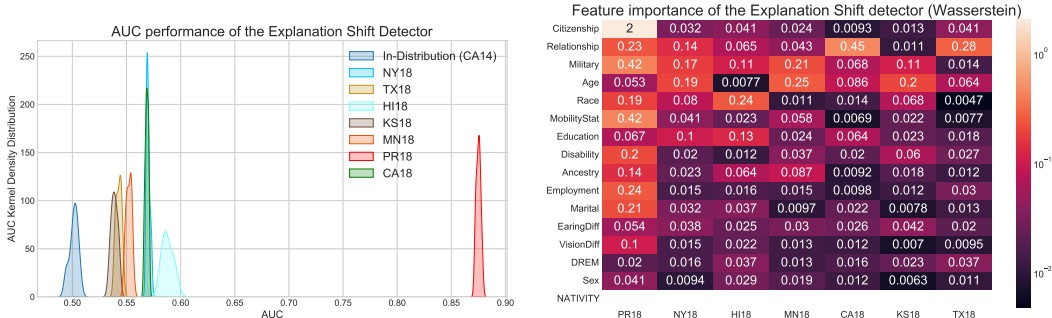

**Figure 5:** The left figure shows a comparison of the performance of the Explanation Shift Detector in different states for the ACS Employment dataset. The right figure shows the feature importance analysis for the same dataset.

Additionally, the feature importance analysis for the same dataset is presented in the right figure in Figure 5.

## D.2  ACS Travel Time

The goal of this task is to predict whether an individual has a commute to work that is longer than $+20$ minutes. For this prediction task, the results are different from the previous two cases; the state with the highest OOD score is $KS18$, with the "Explanation Shift Detector" highlighting features as Place of Birth, Race or Working Hours Per Week. The closest state to ID is CA18, where there is only a temporal shift without any geospatial distribution shift.

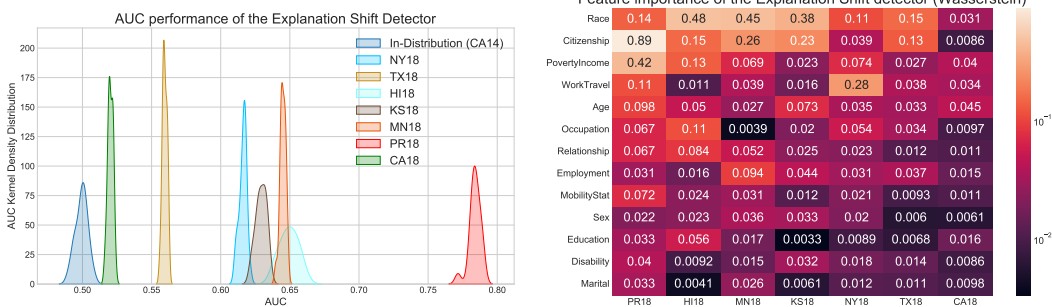

**Figure 6:** In the left figure, comparison of the performance of *Explanation Shift Detector*, in different states for the ACS TravelTime prediction task. In the left figure, we can see how the state with the highest OOD AUC detection is KS18 and not PR18 as in other prediction tasks; this difference with respect to the other prediction task can be attributed to "Place of Birth", whose feature attributions the model finds to be more different than in CA14.

## D.3 ACS Mobility

The objective of this task is to predict whether an individual between the ages of 18 and 35 had the same residential address as a year ago. This filtering is intended to increase the difficulty of the prediction task, as the base rate for staying at the same address is above 90% for the population [54].

The experiment shows a similar pattern to the ACS Income prediction task (cf. Section 4), where the inland US states have an AUC range of $0.55 - 0.70$, while the state of PR18 achieves a higher AUC. For PR18, the model has shifted due to features such as Citizenship, while for the other states, it is Ancestry (Census record of your ancestors' lives with details like where they lived, who they lived with, and what they did for a living) that drives the change in the model.

As depicted in Figure 7, all states, except for PR18, fall below an AUC of explanation shift detection of $0.70$. Protected social attributes, such as Race or Marital status, play an essential role for these states, whereas for PR18, Citizenship is a key feature driving the impact of distribution shift in model.

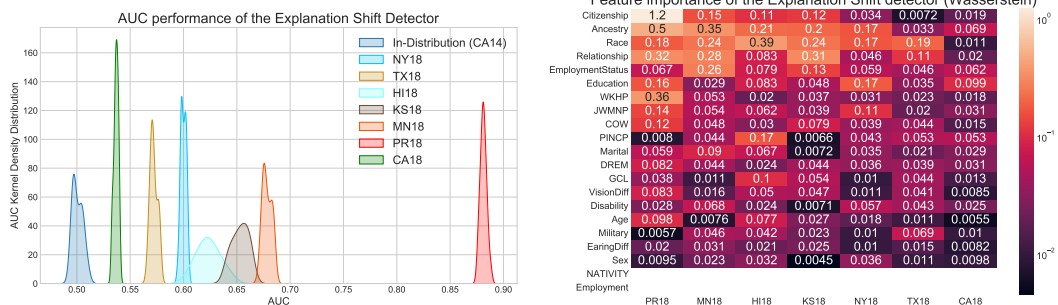

**Figure 7:** Left figure shows a comparison of the *Explanation Shift Detector*'s performance in different states for the ACS Mobility dataset. Except for PR18, all other states fall below an AUC of explanation shift detection of $0.70$. The features driving this difference are Citizenship and Ancestry relationships. For the other states, protected social attributes, such as Race or Marital status, play an important role.

## D.4 StackOverflow Survey Data: Novel Covariate Group

This experimental section evaluates the proposed Explanation Shift Detector approach on real-world data under novel group distribution shifts. In this scenario, a new unseen group appears at the prediction stage, and the ratio of the presence of this unseen group in the new data is varied. The estimator used is a gradient-boosting decision tree or logistic regression, and a logistic regression is used for the detector. The results show that the AUC of the Explanation Shift Detector varies depending on the quantification of OOD explanations, and it show more sensitivity w.r.t. to model variations than other state-of-the-art techniques.

The dataset used is the StackOverflow annual developer survey has over 70,000 responses from over 180 countries examining aspects of the developer experience [55]. The data has high dimensionality, leaving it with $+100$ features after data cleansing and feature engineering. The goal of this task is to predict the total annual compensation.

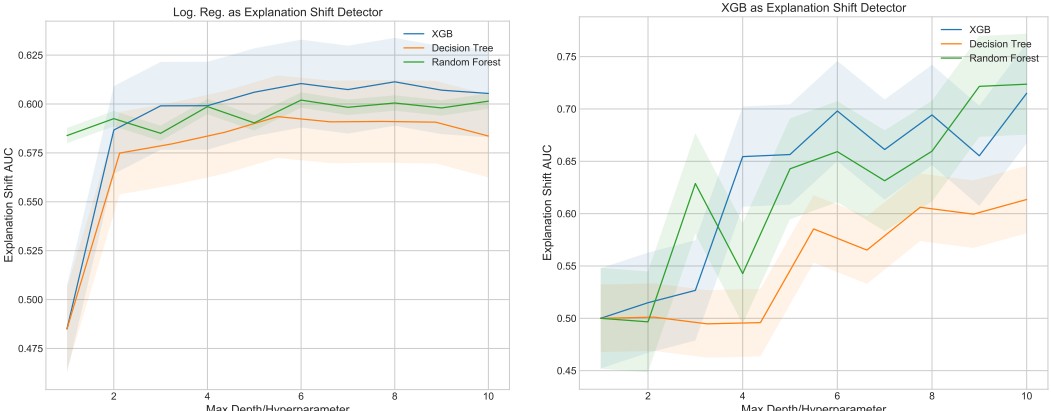

**Figure 8:** Both images represent the AUC of the *Explanation Shift Detector* for different countries on the StackOverflow survey dataset under novel group shift. In the left image, the detector is a logistic regression, and in the right image, a gradient-boosting decision tree classifier. By changing the model, we can see that low-complexity models are unaffected by the distribution shift, while when increasing the model complexity, the out-of-distribution model behaviour starts to be tangible

# E    Experiments with Modeling Methods and Hyperparameters

In the next sections, we are going to show the sensitivity or our method to variations of the estimator $f$, the detector $g$, and the parameters of the estimator $f_\theta$.

As an experimental setup, In the main body of the paper, we have focused on the UCI Adult Income dataset. The experimental setup has been using Gradient Boosting Decision Tree as the original estimator $f_\theta$ and then as "Explanation Shift Detector" $g_\psi$ a logistic regression. In this section, we extend the experimental setup by providing experiments by varying the types of algorithms for a given experimental set-up: the UCI Adult Income dataset using the Novel Covariate Group Shift for the "Asian" group with a fraction ratio of $0.5$ (cf. Section 5).

## E.1    Varying Estimator and Explanation Shift Detector

OOD data detection methods based on input data distributions only depend on the type of detector used, being independent of the estimator. OOD Explanation methods rely on both the model and the data. Using explanations shifts as indicators for measuring distribution shifts impact on the model enables us to account for the influencing factors of the explanation shift. Therefore, in this section, we compare the performance of different types of algorithms for explanation shift detection using the same experimental setup. The results of our experiments show that using Explanation Shift enables us to see differences in the choice of the original estimator $f_\theta$ and the Explanation Shift Detector $g_\phi$

## E.2    Hyperparameters Sensitivity Evaluation

This section presents an extension to our experimental setup where we vary the model complexity by varying the model hyperparameters $\mathcal{S}(f_\theta, X)$. Specifically, we use the UCI Adult Income dataset with the Novel Covariate Group Shift for the "Asian" group with a fraction ratio of $0.5$ as described in Section 5.

In this experiment, we changed the hyperparameters of the original model: for the decision tree, we varied the depth of the tree, while for the gradient-boosting decision, we changed the number of estimators, and for the random forest, both hyperparameters. We calculated the Shapley values using

|                      | Estimator $f_\theta$ |         |       |       |             |          |       |
| Detector $g_\phi$    | XGB   | Log.Reg | Lasso | Ridge | Rand.Forest | Dec.Tree | MLP   |
| -------------------- | ----- | ------- | ----- | ----- | ----------- | -------- | ----- |
| **XGB**              | 0.583 | 0.619   | 0.596 | 0.586 | 0.558       | 0.522    | 0.597 |
| **LogisticReg.**     | 0.605 | 0.609   | 0.583 | 0.625 | 0.578       | 0.551    | 0.605 |
| **Lasso**            | 0.599 | 0.572   | 0.551 | 0.595 | 0.557       | 0.541    | 0.596 |
| **Ridge**            | 0.606 | 0.61    | 0.588 | 0.624 | 0.564       | 0.549    | 0.616 |
| **RandomForest**     | 0.586 | 0.607   | 0.574 | 0.612 | 0.566       | 0.537    | 0.611 |
| **DecisionTree**     | 0.546 | 0.56    | 0.559 | 0.569 | 0.543       | 0.52     | 0.569 |

**Table 6:** Comparison of explanation shift detection performance, measured by AUC, for different combinations of explanation shift detectors and estimators on the UCI Adult Income dataset using the Novel Covariate Group Shift for the "Asian" group with a fraction ratio of 0.5 (cf. Section 5). The table shows that the choice of detector and estimator can impact the OOD explanation performance. We can see how, for the same detector, different estimators flag different OOD explanations performance. On the other side, for the same estimators, different detectors achieve different results.

TreeExplainer [38]. For the Detector choice of model, we compare Logistic Regression and XGBoost models.

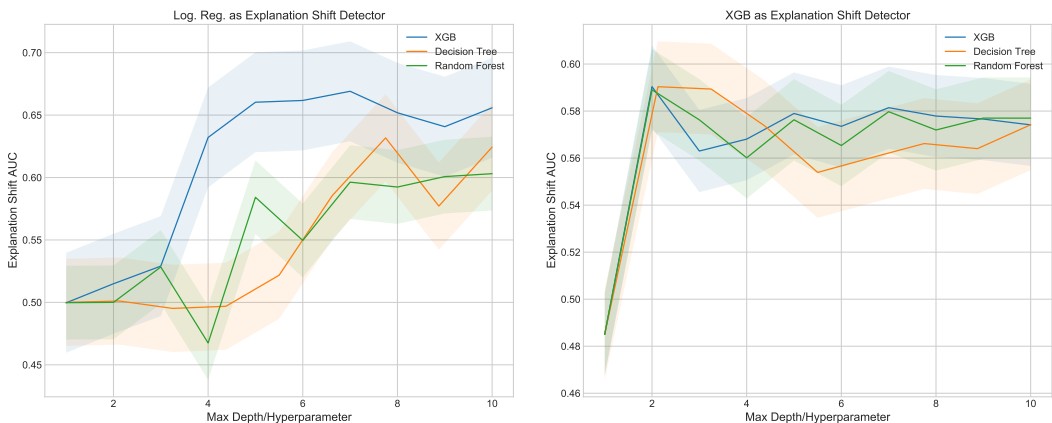

**Figure 9:** Both images represent the AUC of the *Explanation Shift Detector*, in different states for the ACS Income dataset under novel group shift. In the left image, the detector is a logistic regression, and in the right image, a gradient-boosting decision tree classifier. By changing the model, we can see that vanilla models (decision tree with depth 1 or 2) are unaffected by the distribution shift, while when increasing the model complexity, the out-of-distribution impact of the data in the model starts to be tangible

The results presented in Figure 9 show the AUC of the *Explanation Shift Detector* for the ACS Income dataset under novel group shift. We observe that the distribution shift does not affect very simplistic models, such as decision trees with depths 1 or 2. However, as we increase the model complexity, the out-of-distribution data impact on the model becomes more pronounced. Furthermore, when we compare the performance of the *Explanation Shift Detector* across different models, such as Logistic Regression and Gradient Boosting Decision Tree, we observe distinct differences(note that the y-axis takes different values).

In conclusion, the explanation distributions serve as a projection of the data and model sensitive to what the model has learned. The results demonstrate the importance of considering model complexity under distribution shifts.

# F  LIME as an Alternative Explanation Method

Another feature attribution technique that satisfies the aforementioned properties (efficiency and uninformative features Section 2) and can be used to create the explanation distributions is LIME (Local Interpretable Model-Agnostic Explanations). The intuition behind LIME is to create a local interpretable model that approximates the behavior of the original model in a small neighbourhood of the desired data to explain [48, 49] whose mathematical intuition is very similar to the Taylor series.

In this work, we have proposed explanation shifts as a key indicator for investigating the impact of distribution shifts on ML models. In this section, we compare the explanation distributions composed by SHAP and LIME methods. LIME can potentially suffers several drawbacks:

- **Computationally Expensive:** Its currently implementation is more computationally expensive than current SHAP implementations such as TreeSHAP [38], Data SHAP [72, 73] or Local and Connected SHAP [74], the problem increases when we produce explanations of distributions. Even though implementations might be improved, LIME requires sampling data and fitting a linear model which is a computationally more expensive approach than the aforementioned model-specific approaches to SHAP.

- **Local Neighborhood:** The definition of a local "neighborhood", which can lead to instability of the explanations. Slight variations of this explanation hyperparameter lead to different local explanations. In [75] the authors showed that the explanations of two very close points can vary greatly.

- **Dimensionality:** LIME requires as a hyperparameter the number of features to use for the local linear approximation. This creates a dimensionality problem as for our method to work, the explanation distributions have to be from the exact same dimensions as the input data. Reducing the number of features to be explained might improve the computational burden.

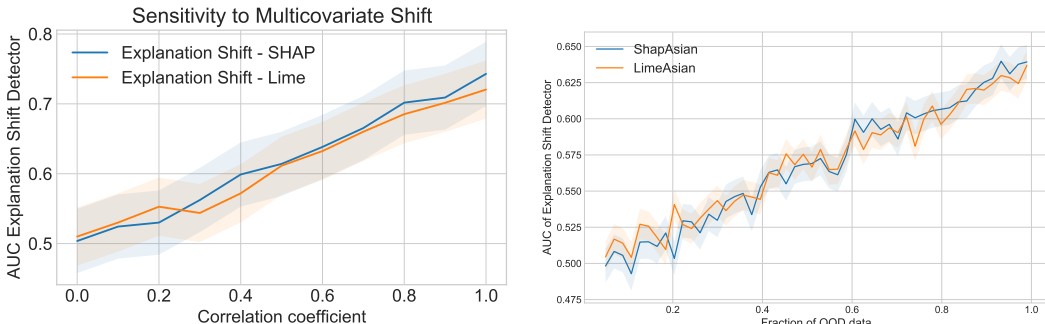

**Figure 10:** Comparison of the explanation distribution generated by LIME and SHAP. The left plot shows the sensitivity of the predicted probabilities to multicovariate changes using the synthetic data experimental setup of 2 on the main body of the paper. The right plot shows the distribution of explanation shifts for a New Covariate Category shift (Asian) in the ASC Income dataset.

Figure 10 compares the explanation distributions generated by LIME and SHAP. The left plot shows the sensitivity of the predicted probabilities to multicovariate changes using the synthetic data experimental setup from Figure 2 in the main body of the paper. The right plot shows the distribution of explanation shifts for a New Covariate Category shift (Asian) in the ASC Income dataset. The performance of OOD explanations detection is similar between the two methods, but LIME suffers from two drawbacks: its theoretical properties rely on the definition of a local neighborhood, which can lead to unstable explanations (false positives or false negatives on explanation shift detection), and its computational runtime required is much higher than that of SHAP (see experiments below).

## F.1 Runtime

We conducted an analysis of the runtimes of generating the explanation distributions using the two proposed methods. The experiments were run on a server with 4 vCPUs and 32 GB of RAM. We used `shap` version $0.41.0$ and `lime` version $0.2.0.1$ as software packages. In order to define the local neighborhood for both methods in this example we use all the data provided as background data. As an estimator, we use an xgboost and compare the results of TreeShap against LIME. When varying the number of samples we use 5 features and while varying the number of features we use $1000$ samples.

Figure 11, shows the wall time required for generating explanation distributions using SHAP and LIME with varying numbers of samples and columns. The runtime required of generating an explanation distributions using LIME is much higher than using SHAP, especially when producing

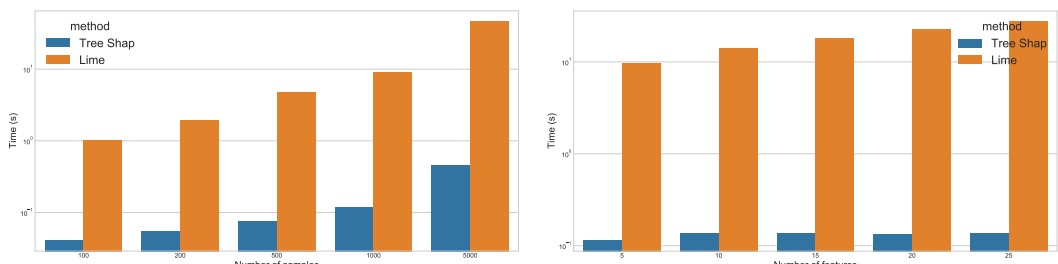

**Figure 11:** Wall time for generating explanation distributions using SHAP and LIME with different numbers of samples (left) and different numbers of columns (right). Note that the y-scale is logarithmic. The experiments were run on a server with 4 vCPUs and 32 GB of RAM. The runtime required to create an explanation distributions with LIME is far greater than SHAP for a gradient-boosting decision tree

explanations for distributions. This is due to the fact that LIME requires training a local model for each instance of the input data to be explained, which can be computationally expensive. In contrast, SHAP relies on heuristic approximations to estimate the feature attribution with no need to train a model for each instance. The results illustrate that this difference in computational runtime becomes more pronounced as the number of samples and columns increases.

We note that the computational burden of generating the explanation distributions can be further reduced by limiting the number of features to be explained, as this reduces the dimensionality of the explanation distributions, but this will inhibit the quality of the explanation shift detection as it won't be able to detect changes on the distribution shift that impact model on those features.

Given the current state-of-the-art of software packages we have used SHAP values due to lower runtime required and that theoretical guarantees hold with the implementations. In the experiments performed in this paper, we are dealing with a medium-scaled dataset with around $\sim 1,000,000$ samples and $20-25$ features. Further work can be envisioned on developing novel mathematical analysis and software that study under which conditions which method is more suitable.

