# OpenReview forum: "Explanation Shift: How Did Distribution Shift Impact the Model?"
_NeurIPS.cc/2023/Conference — Submitted to NeurIPS 2023_

### Official Review · Reviewer_dpht · 2023-06-11

**Soundness:** 2 fair
**Presentation:** 2 fair
**Contribution:** 2 fair
**Rating:** 5
**Confidence:** 3

**Summary:**

This paper introduces a new concept called "explanation shift" for detecting shifts in data distributions with the changes of the attribution distributions on machine learning models. The authors argue that current methods for detecting shifts have limitations in identifying changes in model behavior. Explanation shift provides more sensitive and explainable indicators for these changes. The paper also compares the proposed method against other methods for detecting distribution shifts in both synthetic and real datasets.

**Strengths:**

+ Leveraging the changes of explanations as a manner of detecting the distribution shift is a novel idea.
+ The authors provide a compreshensive analysis to show the connections between explanation shift and various distribution shifts, which could be helpful for readers to understand how to use explanation shift to detect distribution shift

**Weaknesses:**

+ The overall presentation is not clear and many key terminologies and notations are not well explained or defined. For example, in Equation (3), what is $x^*$? What is the formal definition of $S(f_{\theta},x)$? What are the definitions of "sensitivity" and "accountability" which are used as evaluation metrics in Experiments? The lack of clear presentations of these terms makes me extremely hard to understand the key information in this paper
+ Although the authors proposed a new concept called "explanation shift", the technical contribution is still very limited. First of all, the method proposed for detecting the explanation shift (i.e., Section 3) is very simple. But the authors failed to justify why this is an effective method from the theoretical perspective by comparing it against other methods.
+  Some key empirical studies are missing. In Section 5.3, the authors evaluate their methods on some real datasets to detect novel group distribution shift and geopolitical and temporal shift. However, the authors did not perform the same experiments by using baseline methods. Thus it is unclear whether those baseline methods can discover the same types of shifts or not. If yes, then what are the benefits of the proposed method? If not, why are those baseline methods unable to find out those shifts?

**Questions:**

+ Some notations and terminologies should be clearly defined. For example,  in Equation (3), what is $x^*$? What is the formal definition of $S(f_{\theta},x)$? What are the definitions of "sensitivity" and "accountability" which are used as evaluation metrics in Experiments?
+ In terms of the experiments on the real datasets, can the baseline methods discover the same types of shifts?

**Limitations:**

Not applicable.

---

> ### Author Rebuttal · Authors · 2023-08-08
>
> > Some key empirical studies are missing. In Section 5.3, the authors evaluate their methods on some real datasets to detect novel group distribution shift and geopolitical and temporal shift. However, the authors did not perform the same experiments by using baseline methods. Thus it is unclear whether those baseline methods can discover the same types of shifts or not. If yes, then what are the benefits of the proposed method? If not, why are those baseline methods unable to find out those shifts?
>
>
> Our main contribution is the use and analysis of explanation distributions. The classifier with the two-sample test is not novel per se; it represents the application of a two-sample test to the distribution of explanations.
>
> As a baseline comparison method, we use C2ST on the input data distributions as it has been used in Lopez et al. and C2ST on the predictions.
> The result of the baseline comparison method are: (1) compared to input data, that explanations incorporate information about the model and (2) compared to predictions, the distribution of explanations turns out to be more sensitive as the number of dimensions for the same information is higher, and thus, more precise shifts on the data can be detected earlier.
>
> Both baseline studies are derived through mathematical analysis (Section 4), analytical examples (Appendix B), synthetic data (appendix C), Real Data (Section 5 and Appendix), varying Models and Hyperparameters (Appendix E) varying explanations(Appendix F).
> The question of the paper is not just about how baseline methods can detect types of shift, it is about how they relate to the model. In the analytical examples in the Appendix, one can see the mathematical derivations, and in Appendix C, experiments on synthetic data evaluating against measures in other distributions
>
> > Although the authors proposed a new concept called "explanation shift", the technical contribution is still very limited. First of all, the method proposed for detecting the explanation shift (i.e., Section 3) is very simple. But the authors failed to justify why this is an effective method from the theoretical perspective by comparing it against other methods.
>
> Complementing the answer above. The contribution of the work is the conceptualization of a shift in the distribution of explanations to measure changes in the model, we have compared the same simple methodology to other distributions: input data and predictions.  The comparisons are done from a mathematical analysis perspective, deriving from cases where Shapley values can be analytically calculated (Appendix B), Experiments on synthetic data (Appendix C) and different analysis on real data L229-235
>
>
> > Some notations and terminologies should be clearly defined. For example, in Equation (3), what is x*?
>
> Thank you for flagging terminology inconsistency. We will unify. x* is the instance to be predicted, it may be clearer without the *
>
> > What is the formal definition of  S(f,x)?
>
> $S_j(f;x)$ is the short notion for $S_j(val_{f,x})$. "$S_j(f;x)$ stands for the SHAP value of the j'th feature of input x based on model f. We will make our formal description more precise."
> There is also a formal definition 3.1 L52 Does this clarify?
>
> >What are the definitions of "sensitivity" and "accountability" which are used as evaluation metrics in Experiments?
>
> Accountability in Table 1 “We evaluate accountability by checking if the feature attributions of the detection method correspond with the synthetic shift generated in both scenarios”.
>
> Sensitivity: reactivity to measuring the interaction between a distribution and the model. We will clarify this in the manuscript. Thanks for pointing it out.
>
> > In terms of the experiments on the real datasets, can the baseline methods discover the same types of shifts?
>
> In L79 we add a disclaimer, “In practice, multiple types of shifts co-occur together, and their disentangling may constitute a significant challenge that we do not address here”.

---

> > ### Comment · Reviewer_dpht · 2023-08-18
> >
> > Thanks very much for the authors' efforts in providing such detailed answers. I have read other reviewers' comments. I do feel that the authors do need to compare the proposed solution against the baseline methods in literature such as NDCG as mentioned by the reviewer jhhk. So I would maintain my score unless there are more experimental results from the authors.

---

> > > ### Author Response · Authors · 2023-08-21
> > >
> > > Many thanks for the reviewing. We have added a comparison against NDCG in the general comments.

---

> > > > ### Comment · Reviewer_dpht · 2023-08-21
> > > >
> > > > Thanks very much for the authors' efforts! I determined to raise my score from 4 to 5.

---

### Official Review · Reviewer_YKxz · 2023-07-06

**Soundness:** 2 fair
**Presentation:** 2 fair
**Contribution:** 3 good
**Rating:** 5
**Confidence:** 3

**Summary:**

This paper uses explanation shift as a way to detect different types of distribution shift between the training set and unseen (test) data sets. The method is based on measuring the changes between the explanation provided by an explanation approach such as Shapley values, for the two data sets for a trained model. As such, the two data sets could be statistically similar but appear different from the model’s perspective. Overall, the proposed approach is novel and interesting but the paper needs to be improved.

**Strengths:**

To the best of my knowledge, this is a novel approach that uses explanation to detection distribution shift. The proposed method is clear and the method seems to be effective in practice.

**Weaknesses:**

Section 4.1 provides examples where the proposed method works but simple distribution shift evaluation fails. But this does not provide any guarantee whether in general the proposed model is better or not. The same is true for Section 4.3. Section 4.2. provides a disposition but as mentioned by the authors, the prediction shift implies explanation shift, but the opposite is not true. Thus, no conclusion can be ae when there is an explanation shift.

Even though the authors compared their proposed model with the baselines on the synthetic data set in Section 5.1, they have not done it using any real data sets. The real data set is mainly used to study the sensitivity of the model on the parameters.

There is lack of consistency in notation used in the paper that makes it more difficult to follow. Notation changes from one section to another, and in some extreme cases from one example to another. Here are some instances:
1-	Val function is defined differently in Equation 1 and 2.
2-	Equation 3 is not clear and not explained either. What is the expected value is defined on? If it is X, why the notation differs from Equation 2?
3-	There is a sign used in Example 4.2 which is not defined.


The paper benefits from a round of proof-reading.
Line 143: out approach –> our approach
Line 182: a hard tasks --> a hard task (the sentence that includes this is also not clear and needs explanation)
Line 279: AppendixE.1 --> Appendix E.1


**Questions:**

Check the Weaknesses.

**Limitations:**

No.

---

> ### Author Rebuttal · Authors · 2023-08-08
>
>
> > Section 4.1 provides examples where the proposed method works but simple distribution shift evaluation fails. But this does not provide any guarantee whether in general the proposed model is better or not. The same is true for Section 4.3. Section 4.2. provides a disposition but as mentioned by the authors, the prediction shift implies explanation shift, but the opposite is not true. Thus, no conclusion can be ae when there is an explanation shift.
>
> We want to measure changes in the distribution shift and their interaction with the model. Section 4.1. implies that prediction shift is not a reliable measure.
>
> In sections 4.2   we state that shifts in input data are don't relate necessarily to changes on the model model and in section 4.3 that changes in the predictions don't necessarily measure distribution shifts that impact the model.
>
> The mathematical examples aim to showcase situations where explanation shift achieves the desired result against input data shift or prediction shift. It aims to be a conceptual analysis that supports the later experimental one.
>
> The experimental section aims to showcase in which situations is better, for this we have shown analytical, synthetic, and real data experiments. An example is that we have shown how varying hyperparameters of the model  (Appendix E) affect explanation shift where distribution shift is not affected. We also perform experiments varying estimator (f). See L229 – 235.
>
> In summary, the difference with input data shift is that explanation shift relies on the model, and for prediction shift, the higher number of dimensions containing the same information.
>
> Explanation shift implies prediction shift but the other direction does not hold, helping us to provide theoretical evidence that many of the current methods that rely on model predictions are actually not reliable for measuring distribution shift that impacts the model.
>
> > There is lack of consistency in notation used in the paper that makes it more difficult to follow. Notation changes from one section to another, and in some extreme cases from one example to another. Here are some instances:
> >Val function is defined differently in Equation 1 and 2.
>
> Eq. 1 shows the Shapley values for an arbitrary value function val, while eq. 2 defines the specific value function of SHAP-values.
>
> > Equation 3 is not clear and not explained either. What is the expected value is defined on? If it is X, why the notation differs from Equation 2?
>
> Here, $S_j(f;x)$ is the short notion for $S_j(val_{f,x})$ often seen in literature. We will add this to Eq. 3
>
> > There is a sign used in Example 4.2, which is not defined.
>
> We are not sure, which sign is unknown to the reviewer. We hope it is one of the following: ~ (is sampled from); ⊥ (is stochastically independent of); × (Cartesian Product)

---

### Official Review · Reviewer_y637 · 2023-07-06

**Soundness:** 4 excellent
**Presentation:** 4 excellent
**Contribution:** 3 good
**Rating:** 7
**Confidence:** 3

**Summary:**

Detecting shifts in data distribution between training and deployment is critical for ensuring models function as intended and operate in their domain of applicability. However, detecting such shifts is challenging. In this paper, the authors propose an approach based on techniques from the explainability literature. They define the concept of explanation shift and introduce an Explanation Shift Detector. They validate their approach on a synthetic data and 4 tabular datasets, demonstrating improved performance over a range of baselines.

**Strengths:**

The paper is well written, the introduction well motivated, and the formalism both precise and easy for the reader to follow. I found the method interesting, and the analysis of explanation shift detailed and informative. I think this work is a meaningful contribution to the literature.


**Weaknesses:**

The experiments were only conducted on several, relatively simple, tabular datasets. Demonstrating the method for another modality would strengthen the paper.

Please see Questions below.


**Questions:**

L115-119 - Related work in explainability. Lundberg et al. is not the only work relating explainability and distributional shift. For example, Crabbe et al (2020) use example-based explanations to detect out-of-distribution samples, while Hinder et al. (2022) use contrasting explanations to explain concept drift.

Crabbé, Jonathan, et al. "Explaining latent representations with a corpus of examples." Advances in Neural Information Processing Systems 34 (2021): 12154-12166.
Hinder, Fabian, et al. "Contrasting explanation of concept drift." 30th European Symposium on Artificial Neural Networks, Computational Intelligence and Machine Learning, ESANN. 2022.

Figure 1 – I think there is a typo in “Explain Explanation Shift Detector” (unclosed bracket).

L238-241 – It would benefit the reader to provide some additional explanation of some of the baseline methods, particularly those which don’t seem to be discussed elsewhere.

L258 – how does Table 1 “show the results of [your] approach”? More generally Table 1 seems like it would fit more naturally with the discussion of related work.

L263,L265 – I think left and right in Figure 2 have been switched.

Figure 2, right – It is stated in the caption that “good indicators should follow a progressive stead positive slope…”, however this is not discussed in the text. I think the discussion would benefit from this, since it might not be immediately obvious to the reader.


**Limitations:**

Yes

---

> ### Author Rebuttal · Authors · 2023-08-08
>
> We would like to thank the reviewer for helpful comments.
>
> >L115-119 - Related work in explainability. Lundberg et al. is not the only work relating explainability and distributional shift. For example, Crabbe et al (2020) use example-based explanations to detect out-of-distribution samples, while Hinder et al. (2022) use contrasting explanations to explain concept drift.
>
> Many thanks for the related work that can help us better position the work. We will definitely add them in Appendix A, where we compare them to more existing works and consider adding them to the main body.
>
>
> > Demonstrating the method for another modality would strengthen the paper.
>
> Applying explanation distributions to other data modalities is not straightforward due dimensionality of the distributions (e.g. NLP). We have stated this limitation of the scope of the paper.
> Extending to other data modalities remains a further avenue of research where we will need novel methods to measure explanation shift.
>
>
>
> –
> Concerning the rest of the comments, we fully agree with the reviewer and will address them adequately.

---

> > ### Comment · Reviewer_y637 · 2023-08-14
> >
> > I have read all reviews and responses from the Authors and I thank the Authors for their responses.
> >
> > I would like to retain my original (positive) evaluation.

---

### Official Review · Reviewer_jhhk · 2023-07-06

**Soundness:** 2 fair
**Presentation:** 2 fair
**Contribution:** 2 fair
**Rating:** 3
**Confidence:** 4

**Summary:**

The submission proposes an approach to improve model monitoring by rather evaluating changes in explanations instead of input features. The authors provide synthetic examples to justify their method and compare it empirically to existing strategies on tabular datasets.

**Strengths:**

 - The paper addresses an important topic as effective model monitoring based on unlabeled data only is a relevant problem.
 - Although rather, simplistic the synthetic examples help to get a rough idea about the potential benefits of explanation monitoring.
 - The authors provide code as well as tutorials on how to apply their method to ensure reproducibility.


**Weaknesses:**

 - The theoretical analysis is extremely limited such that the overall assumptions under which the proposed method can be expected to yield actual benefits are too vague. Also, the basic notations section seems a bit inflated.
 - I think the novelty is limited as well. Monitoring feature attributions instead of input data is not new and is already offered by popular ML service providers. See for instance here the functionality implemented by Google (https://cloud.google.com/vertex-ai/docs/model-monitoring/monitor-explainable-ai). I would have also liked to see such an alternative approach to use explanations for monitoring somewhere included in the experiments.
 - The conducted numerical experiments are not sufficient to demonstrate the benefits of the proposed approach. If only considering tabular data I think including more than 3 actual datasets and 4 prediction tasks is necessary to be convincing. This is especially true for methods where rigorous theoretical analysis is challenging. See also the question below for further suggestions.
 - The evaluation section is hard to follow, and lacks formulation of insights derived from the experimental results, e.g., it is unclear what benefits can be derived from the feature importance in Figure 4. Given the lack of baselines and justification for those explanations, it is also not clear if they represent useful insights into the effect of a distribution shift on the model’s behavior.  Also, Table 1 comes out of nowhere and is not described sufficiently.
 - Given the limited theoretical and empirical investigation the submission does in my opinion not make a significant contribution to the field.


**Questions:**

Why are only observational Shapley values considered? Have there also been conducted experiments based on the interventional Shapley Values? The interventional approach is more applicable in general and has been proven to result in Shapley values that are closer to explaining the true model behavior [1,2]. I would imagine that this should also have an impact on how distribution changes are reflected in the explanations.

Has it also been tried to apply explanations techniques like SHAP and LIME to the two-sample test classifier instead of the presented approach in Figure 4 (right)? This would be much more general.

Have all of the mentioned baseline methods also been applied to the considered real datasets? This is needed for a comprehensive evaluation.

Why are the investigations only limited to tabular data? Explanations are extremely popular for vision models and even Shapley values can also be approximated quite efficiently when basic feed-forward neural networks are used.  Adebayo et al. [3] evaluate a variety of different explanation methods on vision models for in-domain and out-of-domain instances. Their setup is a little different but maybe the authors can get some inspiration regarding similar experiments for explanation shifts on image data.

Explanations are unstable and might change significantly already for minor input perturbations [4] or other desirable sanity checks that are not satisfied [5]. I wonder whether these phenomena also impact the capabilities of explanation monitoring and would have liked to see a corresponding investigation.

[1] Janzing, Dominik, Lenon Minorics, and Patrick Blöbaum. "Feature relevance quantification in explainable AI: A causal problem." International Conference on artificial intelligence and statistics. PMLR, 2020.
[2] Chen, Hugh, et al. "True to the model or true to the data?." arXiv preprint arXiv:2006.16234 (2020).
[3] Adebayo, Julius, et al. "Debugging Tests for Model Explanations." Advances in Neural Information Processing Systems 33 (2020): 700-712.
[4] Alvarez-Melis, David, and Tommi S. Jaakkola. "On the robustness of interpretability methods." arXiv preprint arXiv:1806.08049 (2018).
[5] Adebayo, Julius, et al. "Sanity checks for saliency maps." Advances in neural information processing systems 31 (2018).


**Limitations:**

I appreciate the discussion at the end of the paper that hints at some relevant limitations.

---

> ### Author Rebuttal · Authors · 2023-08-08
>
>
> We would like to thank the reviewer for helpful comments.
>
> > Why are only observational Shapley values considered? Have there also been conducted experiments based on the interventional Shapley Values? The interventional approach is more applicable in general and has been proven to result in Shapley values that are closer to explaining the true model behavior [1,2]. I would imagine that this should also have an impact on how distribution changes are reflected in the explanations.
>
> In our experiments, we found that the statistical differences between observational and interventional don’t relate to statistically significant changes.
> See attached pdf that we will add to the Appendix of the paper.
>
> > Monitoring feature attributions instead of input data is not new and is already offered by popular ML service providers. See for instance here the functionality implemented by Google (https://cloud.google.com/vertex-ai/docs/model-monitoring/monitor-explainable-ai). I would have also liked to see such an alternative approach to use explanations for monitoring somewhere included in the experiments.
>
> While commercial ML service providers may offer monitoring services utilizing explainable AI, the context of our research is different. Our study aims to contribute to the academic community and the field of machine learning research. The distinction lies in the research's methodology and adherence to a principled investigation and academic standards. To the best of our knowledge, our work is the first to use explanation distribution to investigate the relation between distribution shift and ML models. Thus, our research broadens the spectrum of data analytics approaches to distribution shift detection.
>
> Google offers an individual feature monitoring service, which coincides with the research of Lundberg (https://arxiv.org/pdf/1905.04610.pdf), section 2.7.4 “Local model monitoring…” Our work builds on their previous research line (L115) and we propose the usage of explanation distribution and C2ST.  We provide (i) mathematical derivations (ii) synthetic as well as (iii) real-world data examples of differences between shift of explanation distributions vs. shift of input data and vs. shift of model predictions.
>
> Model monitoring, using explanation deviations as a proxy for model performance degradation, in the absence of labelled data, is a particularly challenging task, where no estimator will perform better consistently, particularly on tabular data L309-L316.
>
> We also provide open-source software, allowing the community to use additional data analytics tools complementing those of big tech companies and ML service providers.
>
> > I would have also liked to see such an alternative approach to use explanations for monitoring somewhere included in the experiments. Has it also been tried to apply explanations techniques like SHAP and LIME to the two-sample test classifier instead of the presented approach in Figure 4 (right)?
>
> In Appendix F, “LIME as an Alternative Explanation Method” we have added experiments using LIME. The results performance results are fairly similar for both synthetic and natural data. The biggest difference seems to be in runtime, as wall time increases.
>
> Even though there could be theoretical differences between different SHAP value estimations (e.g., interventional/observational), our empirical analysis shows that there are few to none. LIME, which is a distinct feature attribution method, leads to similar results. Changes to the model hyperparameters (Appendix E.2 - Figure 9) are much more impactful.
> We have selected Shapley values (and not LIME) because of the theoretical properties that Shapley values have; those properties allow us to develop the mathematical analysis (on Appendix B). We also acknowledge the limitations of our approach and have correspondingly described potential future research venues.
>
> > Explanations are unstable and might change significantly already for minor input perturbations [4] or other desirable sanity checks that are not satisfied [5]. I wonder whether these phenomena also impact the capabilities of explanation monitoring and would have liked to see a corresponding investigation.
>
> Even though explanations are unstable, we deal with shifts in explanation distributions, this is more robust than single-instance explanations (local). For example, the related work, [4], shows explanation instability in local explanations (see their Fig. 2 and Fig. 3). For the related work [5], the same effects can be observed, and in this case, the authors focus on image data, where feature attribution methods are less successful than in tabular data.
>
> > Why are the investigations only limited to tabular data?
>
> Our methodology is limited to tabular data. SHAP values are more reliable and stable in tabular data. Extending to Image or Text data is not straightforward (due to local feature dependencies and much higher explanation dimensions). This, however, remains on our list of further research and is out of scope for this paper. We could further clarify this limitation in the discussion.
>
> > Have all of the mentioned baseline methods also been applied to the considered real datasets?
>
> In Appendix D, there are further experiments on real-world data. We will be happy to extend the Appendix if the reviewer thinks it is needed.
> Note that there are also more experiments in the tutorials of the Python package.
>
>
> > The conducted numerical experiments are not sufficient to demonstrate the benefits of the proposed approach. If only considering tabular data I think including more than 3 actual datasets and four prediction tasks is necessary to be convincing.
>
> Besides numerical experiments, we provide mathematical analysis, derivations on simple cases, synthetic experiments and open-source code. The experiments are done across different types of angles on those datasets L229-235

---

> > ### Comment · Reviewer_jhhk · 2023-08-18
> >
> > Thank you for the clarifications. I have carefully read the authors' rebuttal and my main concerns still hold, in particular regarding novelty and lack of experiments. The authors claim there is no academic work on monitoring feature attributions, as offered by different ML service providers. However, Nigenda et al. presented a detailed description of the algorithm implemented in SageMaker as part of a full KDD paper last year (full ref. below). In particular, in section "4.4 Detecting drift in model feature attributions", they describe how they use a Normalized Discounted Cumulative Gain (NDCG) score for comparing the feature attribution rankings of training and distribution shift data. In comparison to this approach, the novelty of the submission is that instead of NDCG, the authors use a two-sample classifier (which also was proposed previously) to process the shifts in feature attributions.
> >
> > I do appreciate the open-source implementation provided by the authors as this is really something of value to the community - but as is, I think this nice piece of software is the main contribution here. For a fully fledged research paper, however, I would expect at the very least a comparison to NDCG.
> >
> > I therefore maintain my score.
> >
> > David Nigenda, Zohar Karnin, Muhammad Bilal Zafar, Raghu Ramesha, Alan Tan, Michele Donini, and Krishnaram Kenthapadi. 2022. Amazon Sage- Maker Model Monitor: A System for Real-Time Insights into Deployed Machine Learning Models. In Proceedings of the 28th ACM SIGKDD Con- ference on Knowledge Discovery and Data Mining (KDD ’22), August 14–18, 2022, Washington, DC, USA. ACM, New York, NY, USA, 11 pages

---

> > > ### Author Response · Authors · 2023-08-19
> > >
> > > Many thanks for pointing out the related work. Also many thanks for appreciating our open source software compared to ML software providers such as Google or Amazon.
> > >
> > > In the less than a page subsection of the mentioned paper, authors propose a similar approach to Lundberg, but quantified using NDCG between the feature attribution change.
> > > The contribution does not provide any further analysis of why the proposed method works.
> > > The scope of the paper is also a bit distinct, focusing on MLOps software for model monitoring and model retraining rather than the techniques used.
> > >
> > >
> > > Some distinctions:
> > >  - They focus on monitoring model performance. Monitoring model deterioration, particularly on tabular data, is an impossible task where no method will achieve consistent optimal results. We have discussed this aspect, and we focus instead on “How did the distribution shift impact the model.”
> > >  - Their work does not provide any mathematical analysis of why it works.
> > >  - Their experimental part is done only on one dataset under synthetic shift.
> > >  - The experiment only uses a Logistic Regression. We compare several types of algorithms.
> > >
> > > A simple example where their method would not work and ours will is a basic monotonous and uniform covariate shift. $X^{new}_j = X^{tr}_j  + 1$  for every feature(j) of the dataset. Their method will find that the feature attribution order is the same (false negative) while ours will be a true positive.
> > >
> > > Also, if there is a univariate shift in the most relevant feature, there is no guarantee that the method will detect it. Similar hold for the less important feature.
> > >
> > > We will extend the experiments of the main body to include this method and the related work. If the reviewers found it necessary, we will happily provide a comparison in Appendix H between the two papers in mathematical analysis,  synthetic data experiment and natural data.
> > >
> > > Even though both papers handle similar issues, our contribution differs in scope, width of analysis, methods, and depth.

---

> > > > ### Author Response · Authors · 2023-08-21
> > > >
> > > > After reviewing the cited research and those who cite the paper mentioned by the reviewer, we found no work that studies the "explanation shift" issue and investigates interactions between models and shifting data distributions.
> > > >
> > > > We thank the reviewer for the discussion, which helped clarify our work.

---

> > > > > ### Author Response · Authors · 2023-08-21
> > > > >
> > > > > We have added in the general comments as the reviewer suggested:
> > > > >  - A comparison against NDCG, under natural shift, synthetic shift, and mathematical analysis, which we plan to add between the main body and a new appendix.
> > > > > - A comparison between observational and interventional SHAP value estimation.
> > > > >
> > > > > Many thanks again to the reviewer for the time invested in our paper. Its very appreciated.

---

### Author Rebuttal · Authors · 2023-08-08

Experiments comparing Interventional vs Observational SHAP value calculations. To be added to the appendix.

---

> ### Author Response · Authors · 2023-08-21
> **Comparison against NDCG**
>
> Following reviewers comments we have included NDCG comparison as the paper of Nigenda et al.
> Experiments are limited due the short period of time (few hours) and space. If the paper is finally accepted we will provide a fully fledge comparison.
>
> Experiments updated at: https://anonymous.4open.science/r/ExplanationShift-812A/README.md
>
> ### Novel Covariate Group
> Updated figure 3.b https://anonymous.4open.science/r/ExplanationShift-812A/images/NewCategoryBenchmark.pdf
>
> We can observe that NDCG reaches a sensitivity limit around 0.6. A good indicator should follow an increasing steady slope.
> The displayed metric is 0.5-NDCG . To be able to compare with AUC, higher means higher explanation shift.
>
>
> ### Sensitivity experiment under gaussian covariate shift.
> Update Figure 2b of the https://anonymous.4open.science/r/ExplanationShift-812A/images/SOTAsensitivity.pdf
>
> We see that NDCG is highly unstable to this type of shift, not being able to follow a steady increasing slope.
>
> ### Synthetic Data Comparison Against NDCG
>
> Evaluating the change in order of features between in-distribution an out-of-distribution,  if new data comes with a uniform monotonous shift, the feature order won't be affected, but explanation shift based on C2ST will detect.
>
> Given a bivariate normal distribution  $D_X^{tr} = (X_1,X_2) \sim  N(1,I)$ where $I$ is an identity matrix of order two. We create a synthetic targets $Y= X_1^2 \cdot X_2 + \epsilon$  and a machine learning models $f_\theta:D_X \rightarrow D_Y$.
> We create the new data by $D_X^{new} = (X_1,X_2) \sim  N(2,I)$ where $I$, which is a shift of  $D^{new} = D^{tr} +1 $
>  Now we compute the SHAP values for $\mathcal{S}(f_\theta,D_X)$ and compare the order of average contributions.
>
> Having drawn $50,000$ samples from both $D_X$ and $D_X^{new}$, in the following Table, we evaluate whether changes distributions of  explanations and in the order of explanations importance are able to detect changes.
>
>  For this, we compare the one-tailed p-values of the Kolmogorov-Smirnov test for explanation shift, and if the order of the average SHAP value of the distribution changes.
>
>  Explanation shift correctly detects the distribution change and the order of explanations feature importance remains unchanged.
>
> | Comparison                                   | **Conclusions** |
> |---------------------------------------------|-----------------|
> | $P(D_X)$, $P(D_X^{new})$              | Distinct    |
> | $P(S(f_\theta(D_X^{new}))$,  $P(S(f_\theta(D_X^{tr}))$ | Distinct    |
> | $P(S_1(f_\theta,D_X^{new})>S_2(f_\theta,D_X^{new})),P(S_1(f_\theta,D_X^{tr})>S_2(f_\theta,D_X^{tr}))$ |Not Distinct        |
>
>
>
> ### Analytical comparison in Linear and IID cases
> **_Multivariate Shift_**
>
> Let $D_X^{tr}= (D_{X_1}^{tr},D_{X_2}^{tr}) ~ N(\[ \mu_{1}  \\ \mu_{1}\],I) $
> and $D_X^{new}(D_{X_1}^{new},D_{X_2}^{new}) ~ N(\[ \mu_{2}  \\ \mu_{2}\],I) $, where the relationship between $\mu_2 = \mu_1+N$ Where N is real number.
>
> We fit a linear model $f_\theta(X_1,X_2) = \gamma + a\cdot X_1 + b \cdot X_2$.
>
> Then even if the distribution of SHAP values are distinct $S_j(f_\theta,D_X^{tr})$ and $S_j(f_\theta,D_X^{new})$, the order between the distributions is not distinct.
>
> If $S_1(f_\theta,D_X^{tr})>S_2(f_\theta,D_X^{tr})$ then $S_1(f_\theta,D_X^{new})>S_2(f_\theta,D_X^{new})$.
> But  $S_1(f_\theta,D_X^{tr})\neq S_1(f_\theta,D_X^{new})$ and  $S_2(f_\theta,D_X^{tr})\neq S_2(f_\theta,D_X^{new})$
>
> ### Update of table 1
> | **Detection Method**               | **Covariate** | **Uninformative** | **Accountability** |
> |-----------------------------------|--------------|-------------------|--------------------|
> | Explanation distribution ($g_\psi$)| ✔️            | ✔️                | ✔️                 |
> | Input distribution($g_\phi$)       | ✔️            | ❌                | ❌                 |
> | Prediction distribution($g_\Upsilon$)| ✔️            | ✔️                | ❌                 |
> | Input KS                           | ❌            | ❌                | ❌                 |
> | Classifier Drift                  | ✔️            | ❌                | ❌                 |
> | Output KS                         | ✔️            | ✔️                | ❌                 |
> | Output Wasserstein                | ✔️            | ✔️                | ❌                 |
> | Uncertainty                       | ~            | ✔️                | ✔️                 |
> |NDCG.                              | ❌            | ✔️                | ❌                 |
>
>
> ## Conclusion
> In natural data, under novel covariate shift, NDCG is not able to detect upon a threshold of 0.6 fraction data from a previously unseen group.
>
> On synthetic data, NDCG fails to correctly and consistently estimate the multicovariate shift.
>
> Both analytically and on synthetic data experiments, NDCG is not robust to a basic uniform and monotonous shift $X'  = X+1$

---

### Decision · Program_Chairs · 2023-09-21

**Decision:**

Reject

**Comment:**

Reviewers had mixed opinions of this paper. On the one hand, they appreciated the clarity of presentation, and the conceptual contribution of the work. They found the idea of using explanation drift to understand distribution shift novel and interesting. On the other hand, the paper evaluates its method on a rather simple set of benchmarks, and did not compare to a method (NDCG) that tries to accomplish the same thing as their method. The authors conducted a comparison during the rebuttal period, but reviewers did not find the evaluation thorough or convincing enough to change their rating.

For example, one reviewer would have liked to see the authors "compare precision and recall of the AUC score to the NDCG score in settings such as the Novel Group Shift experiment (Fig. 3)," rather than the more qualitative slope comparison done in the rebuttal. Reviewers were also concerned that NDCG was not evaluated on the real-world data.

Overall, I think the paper is not quite complete yet, but if the authors add a more thorough to NDCG and fixing the minor issues raised by reviewers, the paper would be strong enough for publication at a future venue.